# Low-Rank Constraints for Fast Inference in Structured Models

**Justin T. Chiu**[*]
Cornell University
jtc257@cornell.edu

**Yuntian Deng**[*]
Harvard University
dengyuntian@seas.harvard.edu

**Alexander M. Rush**
Cornell University
arush@cornell.edu

## Abstract

Structured distributions, i.e. distributions over combinatorial spaces, are commonly used to learn latent probabilistic representations from observed data. However, scaling these models is bottlenecked by the high computational and memory complexity with respect to the size of the latent representations. Common models such as Hidden Markov Models (HMMs) and Probabilistic Context-Free Grammars (PCFGs) require time and space quadratic and cubic in the number of hidden states respectively. This work demonstrates a simple approach to reduce the computational and memory complexity of a large class of structured models. We show that by viewing the central inference step as a matrix-vector product and using a low-rank constraint, we can trade off model expressivity and speed via the rank. Experiments with neural parameterized structured models for language modeling, polyphonic music modeling, unsupervised grammar induction, and video modeling show that our approach matches the accuracy of standard models at large state spaces while providing practical speedups.

## 1 Introduction

When modeling complex sequential spaces, such as sentences, musical scores, or video frames, a key choice is the internal structural representations of the model. A common choice in recent years is to use neural representations [Bengio et al., 2003, Mikolov et al., 2011, Brown et al., 2020, Boulanger-Lewandowski et al., 2012, Huang et al., 2018, Weissenborn et al., 2020] to store a deterministic history. These models yield strong predictive accuracy but their deterministic, continuous forms provide little insight into the intermediate decisions of the model.

Latent structured models provide an alternative approach where complex modeling decisions are broken down into a series of probabilistic steps. Structured models provide a principled framework for reasoning about the probabilistic dependencies between decisions and for computing posterior probabilities. The structure of the decision processes and the ability to answer queries through probabilistic inference afford interpretability and controllability that are lacking in neural models [Koller and Friedman, 2009, Levine, 2018].

Despite the benefits of structured models, the computational complexity of training scales asymptotically much worse than for neural models, as inference, and therefore training, requires marginalizing

---

[*]Equal contribution
Code is available here.

35th Conference on Neural Information Processing Systems (NeurIPS 2021).

over all possible latent structures. For standard general-purpose models like Hidden Markov Models (HMM) and Probabilistic Context-Free Grammars (PCFG), the runtime of inference scales quadratically and cubically in the number of states respectively, which limits the ability to reach a massive scale. Promisingly, recent work has shown that in specific situations these models can be scaled, and that the increased scale results in commensurate improvements in accuracy – without sacrificing the ability to perform exact inference [Dedieu et al., 2019, Chiu and Rush, 2020, Yang et al., 2021].

In this work, we propose an approach for improving the runtime of a large class of structured latent models by introducing a low-rank constraint. We target the family of models where inference can be formulated through a labeled directed hypergraph, which describes a broad class of dynamic-programming based inference [Klein and Manning, 2004, Huang and Chiang, 2005, Zhou et al., 2006, Javidian et al., 2020, Chiang and Riley, 2020]. We show how under low-rank constraints these models allow for more efficient inference. Imposing a low-rank constraint allows for a key step of inference to be rewritten as a fast matrix-vector product. This approach is also inspired by recent advances in computationally efficient neural attention attention [Katharopoulos et al., 2020, Peng et al., 2021, Choromanski et al., 2020], a significantly different task and formulation, that rewrites matrix-vector products as fast low-rank products using approximate kernel techniques.

We evaluate this approach by learning low-rank structured models for the tasks of language modeling, polyphonic music modeling, unsupervised grammar induction, and video modeling. For these tasks we use a variety of models including HMMs, PCFGs, and Hidden Semi-Markov Models (HSMMs). As the application of low-rank constraints is nontrivial in high-dimensional structured models due to reduced expressivity, we demonstrate effective techniques for overcoming several practical challenges of low-rank parameterizations. We find that our approach achieves very similar results to unconstrained models at large state sizes, while the decomposition allows us to greatly increase the speed of inference. Results on HMMs show that we can scale to more than 16,000 states; results on PCFGs achieve a significant perplexity reduction from much larger state spaces compared to past work [Kim et al., 2019]; and results on HSMMs show that our formulation enables scaling to much larger state spaces for continuous emissions [Fried et al., 2020].

## 2   Background: Latent Structure and Hypergraphs

We consider the problem of modeling a sequence of observations $p(x) = p(x_1, \ldots, x_T)$. These observations can range in complexity from the words in a sentence to a series of co-occurring musical notes, or to features of video frames, and may be discrete or continuous. We assume these observations are generated by an unobserved (latent) structured representation $z$, and therefore model the joint $p(x, z)$. The structure may be sequential or hierarchical, such as latent trees, and the set of structures $\mathcal{Z}$ is combinatorial, i.e. exponential in size with respect to the input sentence. In order to train these models on observations, we must optimize the evidence $p(x) = \sum_z p(x, z)$ by marginalizing over $z$. Scaling this marginalization is the focus of this work.

Hypergraphs are a graphical model formalism for structured distributions that admit tractable inference through dynamic programming [Klein and Manning, 2004, Huang and Chiang, 2005, Zhou et al., 2006, Javidian et al., 2020, Chiang and Riley, 2020].[2] A labeled, directed, acyclic hypergraph consists of a set of nodes $\mathcal{V}$, a set of hyperedges $\mathcal{E}$, and a designated root node $S \in \mathcal{V}$. Each node $v \in \mathcal{V}$ has a collection of labels $\mathcal{L}_v$. Each hyperedge $e \in \mathcal{E}$ has a head node $u$ and tuple of tail nodes, $v = (v_1, \ldots, v_{|e|})$, where $|e|$ is the number of tail nodes. For simplicity, we will assume *at most* 2 tail nodes $v_1, v_2$, and unless noted, a fixed label set $\mathcal{L}$ throughout. Each hyperedge $e$ is associated with a score matrix $\Psi_e \in \mathbb{R}^{\mathcal{L} \times \mathcal{L}^{|e|}}$ with a score for all head and tail labels.[3] We use the notation $[\Psi_e]_{z_u, (z_1, z_2)}$ to indicate the score for head label $z_u$ and tail labels $z_1$ and $z_2$. Finally, we assume we have a topological ordering over the edges.

A hypergraph is used to aggregate scores bottom-up through a dynamic programming (belief propagation) algorithm. Algorithm 1 (left) shows the algorithm. It works by filling in a table vector

---

[2]While the formalism is similar to undirected factor graphs, it allows us to represent more complex distributions: notably dependency structures with unknown topologies, such as latent trees.

[3]This formalism can represent inference in both locally and globally normalized models, although we focus on local normalization in this work.

---

**Algorithm 1** Hypergraph marginalization

| [*Scalar Form*] | [*Matrix Form*] |
|---|---|
| **for** $u \leftarrow v_1, v_2$ hyperedge $e$ topologically **do** | **for** $u \leftarrow v$ hyperedge $e$ topologically **do** |
| $\quad$ **for** $z_u \in \mathcal{L}_u$ **do** | $\quad \alpha_u \overset{+}{\leftarrow} \Psi_e \beta_v$ |
| $\quad\quad [\alpha_u]_{z_u} \overset{+}{\leftarrow} \sum_{z_1, z_2} [\Psi_e]_{z_u, (z_1, z_2)}$ | **return** $\alpha_S^\top \mathbf{1}$ |
| $\quad\quad\quad \cdot [\alpha_{v_1}]_{z_1} [\alpha_{v_2}]_{z_2}$ | |
| **return** $\sum_z [\alpha_S]_z$ | |

---

**Algorithm 2** Hypergraph marginalization for HMMs and PCFGs

| [*HMM - Backward*] | [*PCFG - CKY*] |
|---|---|
| **for** $t \leftarrow (t+1)$ in right-to-left order **do** | **for** $(i, k) \leftarrow (i, j), (j, k)$ in span-size order **do** |
| $\quad$ **for** $z_{t+1} \in \mathcal{L}$ **do** | $\quad$ **for** $z_1, z_2 \in \mathcal{L}_{i,j} \times \mathcal{L}_{j,k}$ **do** |
| $\quad\quad [\beta_{t+1}]_{z_{t+1}} = [\alpha_{t+1}]_{z_{t+1}}$ | $\quad\quad [\beta_{i,j,k}]_{(z_1, z_2)} = [\alpha_{i,j}]_{z_1} [\alpha_{j,k}]_{z_2}$ |
| $\quad \alpha_t \overset{+}{\leftarrow} \Psi_t \beta_{t+1}$ | $\quad \alpha_{i,k} \overset{+}{\leftarrow} \Psi \beta_{i,j,k}$ |
| **return** $\alpha_0^\top \mathbf{1}$ | **return** $\alpha_{1,T}^\top \mathbf{1}$ |

---

$\alpha_v \in \mathbb{R}^{\mathcal{L}}$ for each node $v$ in order, and is initialized to 1 at the leaf nodes.[4] It returns the sum over latent structures, $p(x)$. Counting loops, the worst-case runtime complexity is $O(|\mathcal{E}| \times L^{|e^*|+1})$ where $L = |\mathcal{L}|$ is the size of the label set and $|e^*|$ the max hyperedge tail size. Algorithm 1 (right) shows the same algorithm in matrix form by introducing joined tail vectors $\beta_v \in \mathbb{R}^{\mathcal{L}^{|e|}}$ for each group of nodes $v$. Letting $z_v = (z_1, z_2)$, the joined tail vector contains entries $[\beta]_{z_v} = [\alpha_{v_1}]_{z_1} [\alpha_{v_2}]_{z_2}$.

To make this formalism more concrete, we show how hypergraphs can be used for inference in several structured generative models: hidden Markov models, probabilistic context-free grammars, and hidden semi-Markov models. Inference in these examples are instances of the hypergraph algorithm.

**Example: Hidden Markov Models (HMM)** HMMs are discrete latent sequence models defined by the following generative process: first, a sequence of discrete latent states $z = (z_1, \ldots, z_T)$ with state size $L$ are sampled as a Markov chain. Then each state $z_t$ independently emits an observation $x_t$, i.e.

$$p(x, z) = \prod_{t=1}^{T} p(z_t \mid z_{t-1}) \, p(x_t \mid z_t), \tag{1}$$

where $p(z_t \mid z_{t-1})$ is the transition distribution, $p(x_t \mid z_t)$ the emission distribution, and $p(z_1 \mid z_0)$ is the initial distribution with distinguished start symbol $z_0$.

Given a sequence of observations $x = (x_1, \ldots, x_n)$ we can compute $p(x) = \sum_z p(x, z)$ using a labeled directed hypergraph, with single-tailed edges, nodes corresponding to state positions, labels corresponding to states, and emissions probabilities incorporated into the scoring matrices $\Psi$. There are $T$ scoring matrices, $\Psi_t \in \mathbb{R}^{\mathcal{L} \times \mathcal{L}}$, with entries $[\Psi_t]_{z_t, z_{t+1}} = p(z_{t+1}, x_t \mid z_t)$ corresponding to transitions.[5] Algorithm 2 (left) shows the approach. This requires time $O(TL^2)$ and is identical to the backward algorithm for HMMs.[6]

**Example: Context-Free Grammars (CFG)** CFGs are a structured model defined by the 5-tuple $\mathcal{G} = (S, \mathcal{N}, \mathcal{P}, \mathcal{X}, \mathcal{R})$, where $S$ is the distinguished start symbol, $\mathcal{N}$ is a set of nonterminals, $\mathcal{P}$ is a set of preterminals, $\mathcal{X}$ is the token types in the vocabulary, and $\mathcal{R}$ is a set of grammar rules. Production rules for start, nonterminals, and preterminals take the following forms:[7]

$$S \to A, \quad A \in \mathcal{N}; \quad A \to B\,C, \quad B, C \in \mathcal{N} \cup \mathcal{P}; \quad D \to x, \quad D \in \mathcal{P}, x \in \mathcal{X}. \tag{2}$$

A probabilistic context-free grammar (PCFG) additionally has a probability measure on the set of rules. To compute $p(x_1, \ldots, x_T)$ with a hypergraph, we create one node for each contiguous subspan $[i, k)$ in the sentence. Nodes with $i + 1 < k$ have a nonterminal label set $\mathcal{L} = \mathcal{N}$. Nodes with

---

[4]The accumulation of scores is denoted by $\overset{+}{\leftarrow}$. Multiple hyperedges can have the same head node, whose scores must be added together.

[5]The left-most scoring matrix for the HMM has entries $[\Psi_1]_{z_1, z_2} = p(z_2, x_1 \mid z_1) p(z_1 \mid z_0)$.

[6]In the case of HMMs, the table vectors $\alpha_t$ correspond to the backward algorithm's $\beta$ values.

[7]We restrict our attention to grammars in Chomsky normal form.

$i + 1 = k$ have a preterminal label set $\mathcal{L}_{i,i+1} = \mathcal{P}$. The main scoring matrix is $\Psi \in \mathbb{R}^{\mathcal{L} \times \mathcal{L}^2}$, with entries $[\Psi]_{z_u,(z_1,z_2)} = p(z_1, z_2 \mid z_u)$.[8] Algorithm 2 (right) shows how for every hyperedge we join the scores from the two tail nodes in $\alpha_{i,j}$ and $\alpha_{j,k}$ into joined tail vector $\beta_{i,j,k} \in \mathbb{R}^{\mathcal{L}^2}$. As there are $O(T^3)$ hyperedges and the largest $\mathcal{L}$ is of size $|\mathcal{N}|$, the runtime of the algorithm is $O(T^3 |\mathcal{N}|^3)$. This approach is identical to the CKY algorithm.

**Example: Hidden Semi-Markov Models (HSMM)** HSMMs are extensions of HMMs that allow for generating a variable-length sequence of observations per state. It defines the following generate process: first, we sample a sequence of discrete latent states $z = (z_1, \cdots, z_K)$ with a first-order Markov model. We then use them to generate the length of observations per state. For our experiments we generate independent continuous emissions $x_t$ with a Gaussian distribution for $p(x_i \mid z_k)$. Full details of the inference procedure are given in Appendix E.

# 3 Rank-Constrained Structured Models

For these structured distributions, hypergraphs provide a general method for inference (and therefore training parameterized versions). However, the underlying algorithms scale poorly with the size of the label sets (quadratic for HMM and HSMM, cubic for CFG). This complexity makes it challenging to scale these models and train versions with very large numbers of states.

In this section, we consider an approach for improving the scalability of these models by reducing the dependence of the computational complexity of inference on the label set size. The main idea is to speed up the matrix-vector product step in inference by using a low-rank decomposition of the scoring matrix $\Psi$. In the next section we show that this constraint can be easily incorporated into parameterized versions of these models.

## 3.1 Low-Rank Matrix-Vector Products

The main bottleneck for inference speed is the matrix-vector product $\alpha_u \overset{+}{\leftarrow} \Psi_e \beta_v$ that must be computed for every edge in the hypergraph. As we saw in Algorithm 1 (left), this step takes time $L^{|e|+1}$ to compute, but it can be sped up by making structural assumptions on $\Psi_e$. In particular, we focus on scoring matrices with low rank.

We note the following elementary property of matrix-vector products. If the scoring matrix can be decomposed as the product of two smaller matrices $\Psi_e = U_e V_e^{\top}$, where $U_e \in \mathbb{R}^{\mathcal{L} \times N}$ and $V_e \in \mathbb{R}^{N \times \mathcal{L}^{|e|}}$, then the matrix-vector products can be computed in time $O(|\mathcal{E}| \times L^{|e|} \times N)$ as follows:

$$\Psi_e \beta_v = \left( U_e V_e^{\top} \right) \beta_v = U_e \left( V_e^{\top} \beta_v \right). \tag{3}$$

This reordering of computation exchanges a factor of $L$ for a factor of $N$. When $N \ll L$, this method is both faster and more memory-efficient.

We enforce the low-rank constraint by directly parameterizing the factors $U_e$ and $V_e$ for scoring matrices $\Psi_e$ that we would like to constrain. We treat both $U_e$ and $V_e$ as embedding matrices, where each row corresponds to an embedding of each value of $z_u$ and a joint embedding of $(z_1, z_2)$ respectively:

$$[U_e]_{z_u,n} = c_{z_u} [\phi(f(z_u))]_n \qquad [V_e]_{(z_1,z_2),n} = c_{z_1,z_2} [\phi(g(z_1, z_2))]_n, \tag{4}$$

where $f$ and $g$ are embedding functions; $c_{z_u}$ and $c_{z_1,z_2}$ are constants (used to ensure proper normalization) or clamped potentials (such as conditional probabilities); and $\phi : \mathbb{R}^D \to \mathbb{R}_+^N$ is a function that ensures nonnegativity, necessary for valid probability mass functions. Algorithm 3 shows the role of the low-rank matrix-vector product in marginalization.[9]

---

[8] We have a separate matrix for terminal production on $x$ which we elide for simplicity.

[9] If the normalizing constants are given by $c_{z_u}$, they can be computed from unnormalized $\tilde{U}_e, \tilde{V}_e$ as follows: $c_{z_u} = [\tilde{U}_e \tilde{V}_e^{\top} \mathbf{1}]_{z_u}$ in time $O(L^{|e|} N + LN)$, and similarly for $c_{z_1,z_2}$.

## 3.2 Application to Structured Models

As enforcing a low-rank factorization of every scoring matrix limits the expressivity of a model, we explicitly target scoring matrices that are involved in computational bottlenecks.[10] For these key scoring matrices, we directly parameterize the scoring matrix with a low-rank factorization, which we call a low-rank parameterization. For other computations, we utilize a standard softmax parameterization and do not factorize the resulting scoring matrix. We refer to this as a mixed parameterization.

---
**Algorithm 3** Low-rank marginalization
---
**for** $u \leftarrow v_1, v_2$ hyperedge $e$ topologically **do**
  **for** $n \in 1, \dots, N$ **do**
  $[\gamma]_n = \sum_{z_v} c_v \, [\phi(g(z_1, z_2))]_n \, [\beta_v]_{z_v}$   $\triangleright \, O(L^{|e|})$

  $\alpha_u \overset{+}{\leftarrow} U_e \gamma$   $\triangleright \, O(LN)$
**return** $\alpha_S^\top \mathbf{1}$

---

**Hidden Markov Models** Low-rank HMMs (LHMMs) use the following mixed parameterization, which specifically targets the state-state transition bottleneck by using a low-rank parameterization for the transition distribution, but a softmax parameterization for the emission distribution:

$$p(z_t \mid z_{t-1}) \propto \phi(\mathbf{u}_{z_{t-1}})^\top \phi(\mathbf{v}_{z_t}), \quad p(x_t \mid z_t) \propto \exp(\mathbf{u}_{z_t}^\top \mathbf{v}_{x_t}), \tag{5}$$

where $\mathbf{u}_{z_{t-1}} = f(z_{t-1})$ and $\mathbf{v}_{z_t} = g(z_t)$ are (possibly neural) embedding functions. The parameterizations of the embedding functions $f, g : \mathcal{L} \to \mathbb{R}^D$, as well as the non-negative mapping $\phi : \mathbb{R}^D \to \mathbb{R}^N_+$ are detailed in Appendix F. When performing inference, we treat the emission probabilities $p(x_t \mid z_t)$ as constants, and absorb them into $c_u$.

This allows inference to be run in time $O(TLN)$, where $T$ is the length of a sequence, $L$ the size of the label space, and $N$ the feature dimension.

**Hidden Semi-Markov Models** For low-rank HSMM (LHSMM), we similarly target the transition distribution and keep the standard Gaussian emission distribution:

$$p(z_k \mid z_{k-1}) \propto \mathbf{u}_{z_{k-1}}^\top \mathbf{v}_{z_k}, \quad p(x_t \mid z_k) \propto K_{\text{Gauss}}(\mathbf{u}_{z_k}, \mathbf{x}_t), \tag{6}$$

where $\mathbf{u}_{z_{k-1}} = \phi(f(z_{k-1}))$ and $\mathbf{v}_{z_k} = \phi(g(z_k))$ are state embeddings, while $K_{\text{Gauss}}(\cdot, \cdot)$ is the Gaussian kernel used to model continuous $\mathbf{x}_t$. The full parameterization of the embeddings is given in Appendix F. The total inference complexity is $O(TLMN)$, where $M$ is the maximum length of the observation sequence under any state.

**Context-Free Grammars** For PCFGs, the inference bottleneck is related to the transition from a nonterminal symbol to two nonterminal symbolss ($A \to B\ C$), and we specifically parameterize it using a low-rank parameterization:

$$p(z_{1,N} \mid S) \propto \exp(\mathbf{u}_S^\top \mathbf{u}_{z_{1,N}}), \quad p(z_{i,j}, z_{j,k} \mid z_{i,k}) \propto \begin{cases} \exp(\mathbf{u}_{z_{i,k}}^\top \mathbf{v}_{z_{i,j}\, z_{j,k}}) & \substack{i+1=j \vee \\ j+1=k} \\ \phi(\mathbf{u}'_{z_{i,k}})^\top \phi(\mathbf{v}_{z_{i,j}), z_{j,k}} & \text{o.w.} \end{cases} \tag{7}$$

$$p(x_i \mid z_i) \propto \exp(\mathbf{u}_{z_i}^\top \mathbf{v}_{x_i}),$$

where $\mathbf{u}_z/\mathbf{u}'_z$ is the embedding of $z$ when $z$ is used as head, $\mathbf{v}_x/\mathbf{v}_{z_1,z_2}$ is the embedding of $x/(z_1, z_2)$ when they are used as tail. See Appendix F for the full parameterization, drawn from Kim et al. [2019]. Note that we limit the application of low-rank constraints to nonterminal to nonterminal productions. These productions dominate the runtime as they are applied at $O(T^3)$ hyperedges. This allows inference to be run in time $O(T^3 L^2 N)$, where $T$ is the length of a sequence, $L$ the size of the label space, and $N$ the feature dimension.

# 4 Experimental Setup

We evaluate the application of low-rank constraints with four experiments: sequential language modeling with HMMs, polyphonic music modeling with a large observation space, hierarchical language modelings with PCFGs, and video modeling with HSMMs.

---
[10]For a discussion of the expressivity of low-rank models compared to models with fewer labels, see Appendix A.

**Data** Our first set of experiments evaluate sequential models on PENN TREEBANK dataset (PTB) [Marcus et al., 1993] for the task of word-level language modeling. We use the preprocessing from Mikolov et al. [2011]. The second set of experiments is on polyphonic music modeling [Boulanger-Lewandowski et al., 2012]. We evaluate on four music datasets: Nottingham (Nott), Piano, MuseData (Muse), and JSB chorales (JSB). Each timestep consists of an 88-dimensional binary vector indicating whether a particular note is played. Since multiple notes may be played at the same time, the effective vocabulary size is extremely large. The third set of experiments use PCFGs for language modeling, we also use PTB, but with the splits and preprocessing used in unsupervised constituency parsing [Shen et al., 2018, 2019, Kim et al., 2019]. The last set of experiments use HSMMs for video modeling, where we use CROSSTASK [Zhukov et al., 2019] with 10% of the training data for validation. We follow the preprocessing steps in Fried et al. [2020] and apply PCA to project features to vectors of size 200. For the full details on datasets, please see Appendix D.

**Models and Hyperparameters** For language modeling with HMMs, we experiment with a range of state sizes, $|\mathcal{L}| = L \in \{2^{10}, 2^{11}, 2^{12}, 2^{13}, 2^{14}\}$, and rank $N \in \{L/2, L/4, L/8\}$. For polyphonic music modeling with HMMs, we experiment with states sizes $L \in \{2^7, 2^8, 2^9, 2^{10}, 2^{11}\}$. For language modeling with PCFGs, we use a set of nonterminals of size $|\mathcal{N}| \in \{30, 60, 100\}$ and preterminals of twice the number of nonterminals $|\mathcal{P}| = 2|\mathcal{N}|$. Our smallest setting ($|\mathcal{N}| = 30$, $|\mathcal{P}| = 60$) is the one used in Kim et al. [2019]. For video modeling with HSMMs, we use the same model setting as Fried et al. [2020], but we don't constrain states to the predefined states per task, and we experiment with state sizes $L \in \{2^6, 2^7, 2^8, 2^9, 2^{10}\}$ and rank $N \in \{2^4, 2^5, 2^6, 2^7\}$.

We utilize the feature map $\phi(x) = \exp(Wx)$ for the LHMM and LHSMM, and $\phi(x) = \exp(Wx - \|x\|_2^2/2)$ for the LPCFG. We initialize the parameters of feature maps using orthogonal feature projections [Choromanski et al., 2020], and update it alongside the model parameters. For the full hyperparameter and optimization details, see Appendix G.

**Baselines and Evaluation** The language modeling experiments are evaluated using perplexity. Baselines are neurally parameterized HMM with a standard softmax transition. We also compare to VL-HMM, which makes a strong structural sparsity assumption on the emission distribution [Chiu and Rush, 2020]. We include for reference a state-of-the-art language model, the AWD-LSTM [Merity et al., 2017]. For polyphonic music modeling, we compare our LHMM against RNN-NADE [Boulanger-Lewandowski et al., 2012] which models the full joint distribution of notes as well as temporal dependencies; as well as autoregressive neural models such as the R-Transformer [Wang et al., 2019] (as reported by Song et al. [2019]) and an LSTM (as reported by Ziegler and Rush [2019]); models with latent continuous dynamics such as the LV-RNN [Gu et al., 2015] and SRNN [Fraccaro et al., 2016]; and finally comparable models with latent discrete dynamics, the TSBN [Gan et al., 2015] and the baseline HMM. We evaluate perplexities of our low-rank PCFG (LPCFG) against a softmax PCFG (PCFG) [Kim et al., 2019]. For video modeling, we evaluate the negative log likelihoods on the test set and compare low-rank HSMMs to softmax HSMMs.

# 5 Results

**Hidden Markov Models for Language Modeling** Our main experimental result is that the low-rank models achieve similar accuracy, as measured by perplexity, as our baselines. Fig. 1 shows that perplexity improves as we increase the scale of the HMM, and that the performance of our LHMM also improves at the same rate. At small sizes, the low-rank constraints slightly hinder accuracy; however once the size is large enough, i.e. larger than $2^{12}$, LHMMs with 8:1 state-to-rank ratios perform comparably. [11]

Fig. 1 also contains speed comparisons between HMMs and LHMMs. A state-to-rank ratio of 8:1 matches the accuracy of softmax HMMs at larger state sizes and also gives an empirical speedup of more than 3x at $L = 2^{14}$. As expected, we only see a speedup when the state-to-rank ratio exceeds 2:1, as we replaced the $O(L^2)$ operation with two $O(LN)$ ones. This implies that the low-rank constraint is most effective with scale, where we observe large computational gains at no cost in accuracy.

---

[11]See Appendix H for an analysis of the ranks of HMMs/LHMMs.

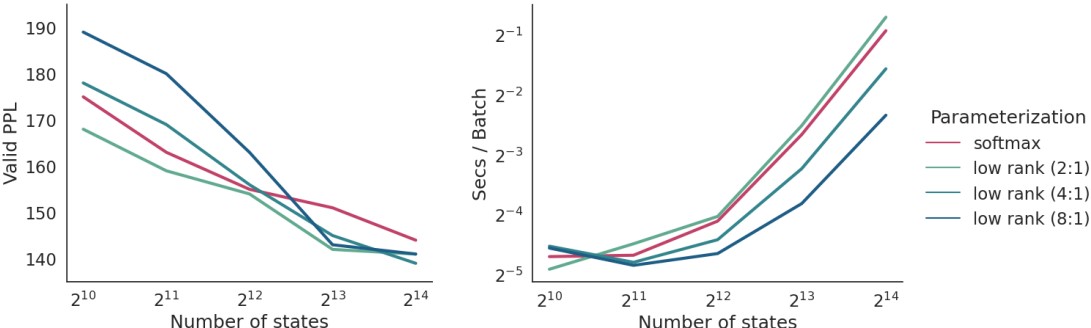

Figure 1: Validation perplexities on PTB versus model scale, as well as speed in seconds per batch.

| Model | Val | Test |
|---|---|---|
| AWD-LSTM | 60.0 | 57.3 |
| VL-HMM | 128.6 | 119.5 |
| HMM | 144.3 | 136.8 |
| LHMM | 141.4 | 131.8 |

| Model | $L : N$ | Train | Val |
|---|---|---|---|
| HMM | - | 95.9 | 144.3 |
| LHMM | 8 | 97.5 | 141.4 |
| LHMM+band | 8 | 101.1 | 143.8 |
| LHMM | 16 | 110.6 | 146.3 |
| LHMM+band | 16 | 96.9 | 138.8 |
| LHMM | 32 | 108.4 | 153.7 |
| LHMM+band | 32 | 110.7 | 145.0 |

Table 1: Model perplexities on PTB. All HMM variants have $L = 2^{14}$ states. (Left): Validation and test perplexities. The LHMM has a state-to-rank ratio $8 : 1$. (Right): Further experiments with extending the low-rank structure of LHMMs with a banded transition structure.

HMMs are outperformed by neural models, and also by VL-HMMs [Chiu and Rush, 2020] which offer similar modeling advantages to HMMs, as shown in Tbl. 1 (left). This indicates that some aspects of performance are not strictly tied to scale. We posit this is due to the problem-specific block-sparse emission constraint in VL-HMMs. While very effective for language modeling, the VL-HMM relies on a hard clustering of states for constraining emissions. This is difficult to apply to problems with richer emission models (as in music and video modeling).

**Hidden Markov Models for Music Modeling** We next apply LHMMs on polyphonic music modeling. This has a max effective vocabulary size of $2^{88}$, as multiple notes may occur simultaneously. Unlike for language modeling, we use a factored Bernoulli emission model, modeling the presence of each note independently. Fig. 2 (right) shows that HMMs are competitive with many of the models on these datasets, including LSTMs. We find that LHMMs achieve performance slightly worse than but comparable to the unconstrained HMMs overall. Fig. 2 (left) shows that the distinction drops with more states. Both HMMs achieve low negative likelihoods (NLL) on the datasets with shorter sequences, Nottingham and JSB, but relatively poorer NLLs on the datasets with longer sequences (Muse and Piano).

**Context-Free Grammars** For syntactic language modeling on PTB, our low-rank PCFG (LPCFG) achieves similar performance to PCFGs, as shown in Table 2 (left), with an improvement in computational complexity. The complexity of inference in PCFGs models is cubic in the number of nonterminals, so even models with $|\mathcal{N}| = 30$ nonterminals are relatively costly. Our approach achieves comparable results with $N = 8$ features. As we scale up the number of nonterminals to $|\mathcal{N}| = 100$, LPCFG stays competitive with a lower computational complexity (since $N < |\mathcal{N}|$). These experiments also demonstrate the importance of scale in syntactic language models with more than 50 point gain in perplexity over a strong starting model.

**CFG Speed** Once the model is large enough, i.e. $|\mathcal{N}| \geq 60$ nonterminals and $|\mathcal{P}| \geq 120$ preterminals, the LPCFG is faster than PCFG, as shown in Tbl. 2 (left). Note that the LPCFG is faster than the CFG even when the number of features $N > \frac{|\mathcal{N}|}{2}$, in contrast to the HMM case where a speedup can only be obtained when $N < L/2$. This is due to the scoring matrix being rectangular: Recall the low-rank matrix product $\Psi\beta = U(V^{\top}\beta)$, where, when specialized to PCFGs, the left-hand side takes

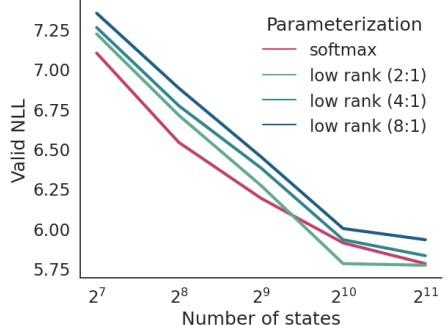

| Model | Nott | Piano | Muse | JSB |
|---|---|---|---|---|
| RNN-NADE | 2.31 | **7.05** | **5.6** | 5.19 |
| R-Transformer | **2.24** | 7.44 | 7.00 | 8.26 |
| LSTM | 3.43 | 7.77 | 7.23 | 8.17 |
| LV-RNN | 2.72 | 7.61 | 6.89 | **3.99** |
| SRNN | 2.94 | 8.20 | 6.28 | 4.74 |
| TSBN | 3.67 | 7.89 | 6.81 | 7.48 |
| HMM | 2.43 | 8.51 | 7.34 | 5.74 |
| LHMM | 2.60 | 8.89 | 7.60 | 5.80 |

Figure 2: Polyphonic music negative log-likelihoods (NLL), measured in nats. (Left): HMM and LHMM validation performance for various state sizes and state:rank ratios. (Right): Test-set NLLs for polyphonic music. The HMM models have $\mathcal{L} = 2^{11}$ states and the LHMM has rank $N = 2^9$, a 4:1 state:rank ratio.

| $|\mathcal{N}|$ | $|\mathcal{P}|$ | Model | $N$ | PPL | Secs |
|---|---|---|---|---|---|
| 30 | 60 | PCFG | - | 252.60 | 0.23 |
| | | LPCFG | 8 | 247.02 | 0.27 |
| | | LPCFG | 16 | 250.59 | 0.27 |
| 60 | 120 | PCFG | - | 234.01 | 0.33 |
| | | LPCFG | 16 | 217.24 | 0.28 |
| | | LPCFG | 32 | 213.81 | 0.30 |
| 100 | 200 | PCFG | - | 191.08 | 1.02 |
| | | LPCFG | 32 | 203.47 | 0.64 |
| | | LPCFG | 64 | 194.25 | 0.81 |

| Model | $L$ | $N$ | NLL | Secs |
|---|---|---|---|---|
| HSMM[12] | 151 | - | 1.432e5 | - |
| HSMM | $2^6$ | - | 1.428e5 | 0.78 |
| HSMM | $2^7$ | - | 1.427e5 | 2.22 |
| HSMM | $2^8$ | - | 1.426e5 | 7.69 |
| LHSMM | $2^7$ | $2^7$ | 1.427e5 | 4.17 |
| LHSMM | $2^8$ | $2^6$ | 1.426e5 | 5.00 |
| LHSMM | $2^9$ | $2^5$ | 1.424e5 | 5.56 |
| LHSMM | $2^{10}$ | $2^4$ | 1.423e5 | 10.0 |

Table 2: (Left): Test perplexities and speeds for PCFG models on PTB. The complexity of PCFG is $O(T^3|\mathcal{N}|^3)$, whereas the complexity of LPCFG is $O(T^3|\mathcal{N}|^2N)$. Speeds are given in seconds per batch. (Right): Negative log likelihoods (NLL) per video and speeds for HSMM models on CROSSTASK. We cannot train HSMMs beyond $2^8$ states due to GPU memory constraints, but we can train LHSMMs with up to $2^{10}$ states. Speeds are given in seconds per batch.

time $O(L^3)$ and the right-hand side takes $O(L^2N + LN)$. For PCFGs, the term $V^\top\beta$ dominates the runtime. This contrasts with HMMs, where both $V^\top\beta$ and the subsequent multiplication by $U$ take the same amount of time, $O(LN)$.

**Hidden Semi-Markov Models for Video Modeling** Table 2 (right) shows the results of video modeling using HSMMs. In addition to using a different hypergraph for inference, these experiments use a continuous Gaussian emission model. By removing the state constraints from tasks, our HSMM baselines get better video-level NLLs than that from Fried et al. [2020] at the cost of more memory consumption. Due to GPU memory constraints, we can only train HSMMs up to $2^8$ states. However, the low-rank parameterization allows models to scale to $2^{10}$ states, yielding an improvement in NLL. Absolute results could likely be improved with more states and by an improved emission parameterization for all models.

**Improving Rank Assumptions** One potential limitation of all low-rank models is that they cannot learn high-rank structures with low $N$. We began to see this issue at a ratio of 16:1 states to features for large HMMs. To explore the effects of this limitation, we perform an additional experiment that combines low-rank features with a sparse component. Specifically we add an efficient high-rank sparse banded transition matrix. The full details are in Appendix I. Tbl. 1 (right) shows that combination with the band structure allows for larger ratios than just the low-rank structure alone, while only adding another operation that costs $O(LN)$.

# 6  Related Work

Similar to our work, other approaches target matrix or tensor operations in inference, and impose structural model constraints to improve computational complexity. Many of the works on HMMs in particular take advantage of the transition structure. The Dense-mostly-constant (DMC) HMM assigns a subset of learnable parameters per row of the transition matrix and sets the rest to a constant, leading to a sub-quadratic runtime [Siddiqi and Moore, 2005]. Other structures have also been explored, such as aligning the states of an HMM to underlying phenomena that allows inference to be sped up [Felzenszwalb et al., 2004, Roweis, 2000]. Additionally, other methods take advantage of emission structure in HMMs in order to scale, such as the Cloned HMM [Dedieu et al., 2019] and VL-HMM [Chiu and Rush, 2020]. Compared to these approaches, our method is more flexible and generic, since it can be applied in a non-application-specific manner, and even extended with high-rank components (such as banded structure).

Low-rank structure has been explored in both HMMs [Siddiqi et al., 2009], a generalization of PCFGs called weighted tree automata [Rabusseau et al., 2015], and conditional random fields [Thai et al., 2018]. The reduced-rank HMM [Siddiqi et al., 2009] has at most 50 states, and relies on spectral methods for training. The low-rank weighted tree automata [Rabusseau et al., 2015] also trains latent tree models via spectral methods. We extend the low-rank assumption to neural parameterizations, which have been shown to be effective for generalization [Kim et al., 2019, Chiu and Rush, 2020], and directly optimize the evidence via gradient descent. Finally, Thai et al. [2018] do not take advantage of the low-rank parameterization of their CRF potentials for faster inference via low-rank matrix products, a missed opportunity. Instead, the low-rank parameterization is used only as a regularizer, with the full potentials instantiated during inference.

Concurrent work in unsupervised parsing uses a tensor decomposition to scale PCFGs to large state spaces [Yang et al., 2021]. Our low-rank decomposition of the flattened head-tail scoring matrix is more general, resulting in worse scaling for the PCFG setting but with wider applicability, as shown by experiments with HMMs and HSMMs.

# 7  Conclusion

This work improves the scaling of structured models by establishing the effectiveness of low-rank constraints for hypergraph models. We show that viewing a key step of inference in structured models as a matrix-vector product, in combination with a low-rank constraint on relevant parameters, allows for an immediate speedup. Low-rank inference allows us to obtain a reduction in the asymptotic complexity of marginalization at the cost of a constrained model. Our approach applies to a wide class of models, including HMMs, HSMMs, and PCFGs. Through our experiments on language, video, and polyphonic music modeling, we demonstrate an effective approach for overcoming the practical difficulty of applying low-rank constraints in high dimensional, structured spaces by targeting and constraining model components that bottleneck computation. Future work includes exploration of other structural constraints for speeding up matrix-vector products [Dao et al., 2020] performed in inference, as well as application to models where exact inference is intractable.

## Acknowledgments and Disclosure of Funding

We thank Nikita Kitaev for the discussion that sparked this project. We thank Sam Wiseman, Jack Morris, and the anonymous reviewers for valuable feedback. Yuntian Deng is sponsored by NSF 1704834, Alexander Rush by NSF CAREER 1845664, and Justin Chiu by an Amazon research award.

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
