# A   Expressivity of Low-Rank Models

We focus on the simplest case of HMMs for an analysis of expressivity. In the case of Gaussian emissions, a model with more states but low rank is more expressive than a model with fewer states because for a single timestep, a larger mixture of Gaussians is more expressive. In the case of discrete emissions, however, the emission distribution for a single timestep (i.e. $\sum_z p(x,z)$) is not more expressive. Instead, we show that there exists joint marginal distributions of discrete $x$ over multiple timesteps that are captured by large state but low-rank HMMs, but not expressible by models with fewer states.

We construct a counter-example with a sequence of length $T = 2$ and emission space of $x_t \in \{0, 1, 2\}$. We show that a 3-state HMM with rank 2, HMM-3-2, with manually chosen transitions and emissions, cannot be modeled by any 2-state HMM. The transition probabilities for the HMM-3-2 are given by (rows $z_t$, columns $z_{t+1}$)

$$p(z_{t+1}|z_t) = \begin{bmatrix} \frac{1}{3} & \frac{1}{3} & \frac{1}{3} \\ 0 & 1 & 0 \\ \frac{1}{2} & 0 & \frac{1}{2} \end{bmatrix} = \begin{bmatrix} \frac{1}{3} & \frac{2}{3} \\ 1 & 0 \\ 0 & 1 \end{bmatrix} \begin{bmatrix} 0 & 1 & 0 \\ \frac{1}{2} & 0 & \frac{1}{2} \end{bmatrix} = UV^T,$$

emission probabilities by (rows $z_t$, columns $x_t$):

$$p(x_t|z_t) = \begin{bmatrix} 1 & 0 & 0 \\ 0 & 1 & 0 \\ 0 & 0 & 1 \end{bmatrix},$$

and starting distribution

$$P(z_1 \mid z_0) = \begin{bmatrix} \frac{1}{3} & \frac{1}{3} & \frac{1}{3} \end{bmatrix}.$$

This yields the following marginal distribution (row $x_1$, column $x_2$):

$$p(x_1, x_2) = \begin{bmatrix} \frac{1}{9} & \frac{1}{9} & \frac{1}{9} \\ 0 & \frac{1}{3} & 0 \\ \frac{1}{6} & 0 & \frac{1}{6} \end{bmatrix}.$$

Next, we show that there does not exist a 2-state HMM that can have this marginal distribution. Assuming the contrary, that there exists a 2-state HMM that has this marginal distribution, we will first show that there is only one possible emission matrix. We will then use that to further show that the posterior, then transitions also must be sparse, resulting in a marginal emission distribution that contradicts the original assumption.

We start by setting up a system of equations. The marginal distribution over observations is obtained by summing over $z_1, z_2$:

$$p(x_1, x_2) = \sum_{z_2} \left( \sum_{z_1} p(x_1, z_1, z_2) \right) p(x_2 \mid z_2).$$

Let the inner term be $f(x_1, z_1) = \sum_{z_1} p(x_1, z_1, z_2) = p(x_1, z_2)$. In a small abuse of notation, let $p(x_2 \mid z_2 = 0)$ be a row vector with entries $[p(x_2 \mid z_2 = 0)]_x = p(x_2 = x \mid z_2 = 0)$, and similarly for $p(x_2 \mid z_2 = 1)$. We then have, first summing over $z_1$,

$$P(x_1, x_2) = \begin{bmatrix} \frac{1}{9} & \frac{1}{9} & \frac{1}{9} \\ 0 & \frac{1}{3} & 0 \\ \frac{1}{6} & 0 & \frac{1}{6} \end{bmatrix} = \begin{bmatrix} f(0,0) & f(0,1) \\ f(1,0) & f(1,1) \\ f(2,0) & f(2,1) \end{bmatrix} \begin{bmatrix} p(x_2|z_2 = 0) \\ p(x_2|z_2 = 1) \end{bmatrix}.$$

We can determine the first row of the emission matrix, $p(x_2 \mid z_2 = 0)$ from the second row of this system of equations, rewritten here:

$$p(x_2 \mid x_1 = 1) = f(1,0)p(x_2|z_2 = 0) + f(1,1)p(x_2|z_2 = 1) = \begin{bmatrix} 0 & \frac{1}{3} & 0 \end{bmatrix}.$$

We can deduce that $f(1,0), f(1,1) > 0$, otherwise $p(x_1, x_2 = 1) = 0 \neq \frac{1}{3}$. Without loss of generality, assume $f(1,0) > 0$, then $p(x_2 = 0|z_2 = 0) = p(x_2 = 2|z_2 = 0) = 0$, since $p(x_1 = 1, x_2 = 0) = p(x_1 = 1, x_2 = 2) = 0$. Therefore,

$$p(x_2|z_2 = 0) = \begin{bmatrix} 0 & 1 & 0 \end{bmatrix}.$$

We can similarly determine the second row of the emission matrix, $p(x_2 \mid z_2 = 1)$, from the last row of the system of equations:

$$p(x_2 \mid x_1 = 2) = f(2,0)p(x_2|z_2 = 0) + f(2,1)p(x_2|z_2 = 1) = \begin{bmatrix} \frac{1}{6} & 0 & \frac{1}{6} \end{bmatrix}.$$

As we determined that $p(x_2|z_2 = 0) = \begin{bmatrix} 0 & 1 & 0 \end{bmatrix}$, $f(2,0)$ must be 0, otherwise $p(x_2 = 1|x_1 = 2) > 0$. Therefore $f(2,1)p(x_2|z_2 = 1) = \begin{bmatrix} \frac{1}{6} & 0 & \frac{1}{6} \end{bmatrix}$, yielding

$$p(x_2|z_2 = 1) = \begin{bmatrix} \frac{1}{2} & 0 & \frac{1}{2} \end{bmatrix}.$$

Putting it together, the full emission matrix is given by

$$p(x_t|z_t) = \begin{bmatrix} 0 & 1 & 0 \\ \frac{1}{2} & 0 & \frac{1}{2} \end{bmatrix}.$$

This allows us to find the posterior distribution $p(z_1 \mid x_1)$ via Bayes' rule:

$$p(z_1 = 1 \mid x_1 = 1) = \frac{p(x_1 = 1 \mid z_1 = 1)p(z_1)}{p(x_1 = 1)} = \frac{0 \cdot p(z_1)}{p(x_1 = 1)} = 0,$$

implying $p(z_1 = 0 \mid x_1 = 1) = 1$. By similar reasoning, we have $p(z_1 = 1 \mid x_1 = 0) = 1$ and $p(z_1 = 1 \mid x_1 = 2) = 1$.

Given the sparse emissions and posteriors, we will show that the transitions must be similarly sparse, resulting in an overly sparse marginal distribution over emissions (contradiction). We can lower bound

$$0 = p(x_2 = 1 \mid x_1 = 2) \geq p(x_2 = 1 \mid z_2 = 0)p(z_2 = 0 \mid z_1 = 1)p(z_1 = 1 \mid x_1 = 2)$$

by the definition of total probability and nonnegativity of probability. Then, substituting $p(x_2 = 1 \mid z_2 = 0) = 1$, we have

$$0 = p(x_2 = 1 \mid x_1 = 2) \geq p(z_2 = 0 \mid z_1 = 1),$$

from which we can deduce $p(z_2 = 0 \mid z_1 = 1) = 0$.

Now, we will show that $p(x_2 = 1 \mid x_1 = 0) = 0$, which contradicts the marginal distribution. We have

$$p(x_2 = 1 \mid x_1 = 0) = \sum_{z_1, z_2} p(x_2 = 1 \mid z_2)p(x_2 \mid z_1)p(z_1 \mid x_1 = 0)$$

$$= p(x_2 = 1 \mid z_2 = 1)p(z_2 = 1 \mid z_1 = 1)p(z_1 = 1 \mid x_1 = 0),$$

where we obtained the second equality because $p(z_1 = 0 \mid x_1 = 0) = 0$ and $p(z_2 = 0 \mid z_1 = 1)$. As $p(x_2 = 1 \mid z_1 = 1) = 0$, we have $p(x_2 = 1 \mid z_1 = 1) = 0 \neq \frac{1}{3}$. As this is a contradiction, we have shown that there exists a marginal distribution modelable with a 3-state HMM with rank 2, but not a 2-state HMM.

# B Low-Rank Hypergraph Marginalization for HMMs and PCFGs

We provide the low-rank hypergraph marginalization algorithms for HMMs and PCFGs in Alg. 4, with loops over labels $z$ (and products of labels) and feature dimensions $n$ left implicit for brevity. We also assume that the label sets for PCFG are uniform for brevity – in practice, this can easily be relaxed (this was not assumed in Alg. 2). We show how the normalizing constants $c$ are explicitly computed using the unnormalized low-rank factors in each algorithm.

**Algorithm 4** Low-rank hypergraph marginalization for HMMs and PCFGs

| | |
|---|---|
| [*HMM - Backward*] | [*PCFG - CKY*] |
| $[\tilde{V}]_{z,n} = [\phi(g(z))]_n$ | $[\tilde{V}]_{z_1,z_2,n} = [\phi(g(z_1,z_2))]_n$ |
| $[\tilde{U}]_{z,n} = [\phi(f(z))]_n$ | $[\tilde{U}]_{z_u,n} = [\phi(f(z_u))]_n$ |
| $[c]_z = [\tilde{U}\tilde{V}^\top \mathbf{1}]_z$ | $[c]_{z_u} = [UV^\top \mathbf{1}]_{z_u}$ |
| **for** $t \leftarrow (t+1)$ in right-to-left order **do** | **for** $(i,k) \leftarrow (i,j),(j,k)$ in span-size order **do** |
| $\quad [\beta_{t+1}]_{z_{t+1}} = [\alpha_{t+1}]_{z_{t+1}}$ | $\quad [\beta_{i,j,k}]_{(z_1,z_2)} = [\alpha_{i,j}]_{z_1}[\alpha_{j,k}]_{z_2}$ |
| $\quad [V_t]_{z_{t+1},n} = [\tilde{V}]_{z_{t+1},n}$ | $\quad [V_{i,j,k}]_{z_1,z_2,n} = [\tilde{V}]_{z_1,z_2,n}$ |
| $\quad [U_t]_{z_t,n} = p(x_t \mid z_t)[c]_{z_t}[\tilde{U}]_{z_t,n}$ | $\quad [U_{i,j,k}]_{z_u,n} = [c]_{z_u}[\tilde{U}]_{z_u,n}$ |
| $\quad \alpha_t \overset{+}{\leftarrow} U_t(V_t^\top \beta_{t+1})$ | $\quad \alpha_{i,k} \overset{+}{\leftarrow} U_{i,j,k}(V_{i,j,k}^\top \beta_{i,j,k})$ |
| **return** $\alpha_0^\top \mathbf{1}$ | **return** $\alpha_{1,T}^\top \mathbf{1}$ |

# C   Extension of the Low-Rank Constraint to Other Semirings

Enforcing low-rank constraints in the scoring matrices $\Psi_e$ leads to a speedup for the key step in the hypergraph marginalization algorithm:

$$\Psi_e \beta_v = \left(U_e V_e^\top\right)\beta_v = U_e\left(V_e^\top \beta_v\right), \tag{8}$$

where $[\beta_v]_{z_1,z_2} = [\alpha_1]_{z_1}[\alpha_2]_{z_2}$. While the low-rank constraint allows for speedups in both the log and probability semirings used for marginal inference, the low-rank constraint does not result in speedups in the tropical semiring, used for MAP inference. To see this, we first review the low-rank speedup in scalar form. The key matrix-vector product step of marginal inference in scalar form is given by

$$\sum_{z_1,z_2}[\Psi_e]_{z_u,(z_1,z_2)}[\beta]_{z_1,z_2} = \sum_{z_1,z_2}\sum_n [U_e]_{z_u,n}[V_e]_{(z_1,z_2),n}[\beta]_{z_1,z_2}$$
$$= \sum_n \sum_{z_1,z_2}[U_e]_{z_u,n}[V_e]_{(z_1,z_2),n}[\beta]_{z_1,z_2}$$
$$= \sum_n [U_e]_{z_u,n}\sum_{z_1,z_2}[V_e]_{(z_1,z_2),n}[\beta]_{z_1,z_2},$$

which must be computed for each $z_u$. The first line takes $O(\mathcal{L}^{|e|+1})$ computation, while the last line takes $O(\mathcal{L}^{|e|}N)$ computation. The speedup comes rearranging the sum over $(z_1, z_2)$ and $n$, then pulling out the $U_e$ factor, thanks to the distributive propery of multiplication. When performing MAP inference instead of marginal inference, we take a max over $(z_1, z_2)$ instead of a sum. Unfortunately, in the case of the max-times semiring used for MAP inference, we cannot rearrange max and sum, preventing low-rank models from obtaining a speedup:

$$\max_{z_1,z_2}[\Psi_e]_{z_u,(z_1,z_2)}[\beta]_{z_1,z_2} = \max_{z_1,z_2}\sum_n [U_e]_{z_u,n}[V_e]_{(z_1,z_2),n}[\beta]_{z_1,z_2}$$
$$\neq \sum_n \max_{z_1,z_2}[U_e]_{z_u,n}[V_e]_{(z_1,z_2),n}[\beta]_{z_1,z_2}.$$

# D   Data Details

For language modeling on PENN TREEBANK (PTB) [Marcus et al., 1993] we use the preprocessing from Mikolov et al. [2011], which lowercases all words and substitutes OOV words with UNKs. The dataset consists of 929k training words, 73k validation words, and 82k test words, with a vocabulary of size 10k. Words outside of the vocabulary are mapped to the UNK token. We insert EOS tokens after each sentence, and model each sentence, including the EOS token, independently.

The four polyphonic music datasets, Nottingham (Nott), Piano, MuseData (Muse), and JSB chorales (JSB), are used with the same splits as Boulanger-Lewandowski et al. [2012]. The data is obtained via

| Dataset | Avg Len | Total Length | | |
| --- | --- | --- | --- | --- |
| | | Train | Valid | Text |
| Nott | 254.4 | 176,551 | 45,513 | 44,463 |
| Piano | 872.5 | 75,911 | 8,540 | 19,036 |
| Muse | 467.9 | 245,202 | 82,755 | 64,339 |
| JSB | 60.3 | 64,339 | 4,602 | 4,725 |

Table 3: The lengths for the four polyphonic music datasets. The average length of an example in the training split for each dataset is given.

the following script. Each timestep consists of an 88-dimensional binary vector indicating whether a particular note is played. Since multiple notes may be played at the same time, the effective vocabulary size is extremely large. The dataset lengths are given in Table 3.

In experiments with PCFGs for language modeling, we also use PTB, but with the splits and preprocessing used in unsupervised constituency parsing [Shen et al., 2018, 2019, Kim et al., 2019]. This preprocessing discards punctuation, lowercases all tokens, and uses the 10k most frequent words as the vocabulary. The splits are as follows: sections 2-21 for training, 22 for validation, 23 for test. Performance is evaluated using perplexity.

In experiments with HSMMs for video modeling, we use the *primary* section of the CROSSTASK dataset [Zhukov et al., 2019], consisting of about 2.7k instructional videos from 18 different tasks such as "Make Banana Ice Cream" or "Change a Tire". We use the preprocessing from Fried et al. [2020], where pretrained convolutional neural networks are first applied to extract continuous image and audio features for each frame, followed by PCA to project features to 300 dimensions.[13] We set aside 10% of the training videos for validation.

# E    Generative Process of HSMM

We use an HSMM to model the generative process of the sequence of continuous features for each video. The HSMM defines the following generative process: first, we sample a sequence of discrete latent states $z = (z_1, \cdots, z_K)$ with a first-order Markov model. Next, we sample the length of observations under each state from a Poisson distribution $l_k \sim \text{Poisson}(\lambda_{z_k})$ truncated at max length $M$. The joint distribution is defined as

$$p(x, z, l) = \prod_{k=1}^{K} p(z_k \mid z_{k-1}) \, p(l_k \mid z_k) \prod_{i=l_1+\cdots+l_{k-1}}^{l_1+\cdots+l_k} p(x_i \mid z_k), \tag{9}$$

where the sequence length $T$ can be computed as $T = \sum_{k=1}^{K} l_k$. In this work, we only consider modeling continuous $x_t$, so we use a Gaussian distribution for $p(x_i \mid z_k)$.

To compute $p(x)$, we can marginalize $l, z$ using dynamic programming similar to HMMs, except that we have an additional factor of $M$: the overall complexity is $O(T \times M \times L^2)$ (ignoring the emission part since they are usually not the bottleneck). We refer to Yu [2010] for more details.

# F    Full Parameterization of HMMs, PCFGs, and HSMMs

In this section, we present more details on the parameterizations of the HMM, PCFG, and HSMM. The main detail is where and how are neural networks used to parameterize state representations.

---

[13]https://github.com/dpfried/action-segmentation

For low-rank HMMs (LHMMs) we use the following mixed parameterization that specifically targets the state-state bottleneck:

$$p(z_1 \mid z_0) \propto \phi(f_1(\mathbf{u}_{z_0}))^\top \phi(\mathbf{v}_{z_1})$$
$$p(z_t \mid z_{t-1}) \propto \phi(\mathbf{u}_{z_{t-1}})^\top \phi(\mathbf{v}_{z_t}) \tag{10}$$
$$p(x_t \mid z_t) \propto \exp(\mathbf{u}_{z_t}^\top f_2(\mathbf{v}_{x_t})),$$

where $\mathbf{u}_z$ is the embedding of $z$ when $z$ is used as head, $\mathbf{v}_z$ its embedding when used as tail, $f_1, f_2$ are MLPs with two residual layers, and feature map $\phi(x) = \exp(Wx)$.

The PCFG uses a similar mixed parameterization. These probabilities correspond to start ($S \to A$), preterminal ($T \to x$), and standard productions ($A \to B\ C$) respectively.

$$p(z_{1,N} \mid S) \propto \exp(f_1(\mathbf{u}_S)^\top \mathbf{u}_{z_{1,N}})$$
$$p(x_i \mid z_i) \propto \exp(\mathbf{u}_{z_i}^\top f_2(\mathbf{v}_{x_i}))$$
$$p(z_{i,j}, z_{j,k} \mid z_{i,k}) \propto \begin{cases} \exp(\mathbf{u}_{z_{i,k}}^\top \mathbf{v}_{z_{i,j}, z_{j,k}}) & \substack{i+1=j\ \vee \\ j+1=k} \\ \phi(\mathbf{u}'_{z_{i,k}})^\top \phi(\mathbf{v}_{z_{i,j}, z_{j,k}}) & \text{o.w.} \end{cases} \tag{11}$$

where $\mathbf{u}_z/\mathbf{u}'_z$ is the embedding of $z$ when $z$ is used as head, $\mathbf{v}_x/\mathbf{v}_{z_1,z_2}$ is the embedding of $x/(z_1, z_2)$ when they are used as tail, and $f_1, f_2$ are MLPs with two residual layers as in Kim et al. [2019]. We use the feature map $\phi(x) = \exp(Wx - \|x\|_2^2/2)$.

For both HMMs and neural PCFG models, we use the same parameterization of the MLPs $f_1$ and $f_2$ as Kim et al. [2019]:

$$f_i(x) = g_{i,1}(g_{i,2}(W_i x)),$$
$$g_{i,j}(y) = \text{ReLU}(U_{i,j}\text{ReLU}(V_{i,j}y)) + y, \tag{12}$$

with $i, j \in \{1, 2\}$, and $W_i, V_{i,j}, U_{i,j} \in \mathbb{R}^{D \times D}$.

For HSMMs, the baseline (HSMM in Table 2) follows the fully unsupervised setting in Fried et al. [2020] except that we don't apply any state constraints from the prior knowledge of each task.[14] The model maintains a log transition probability lookup table for $p(z_k \mid z_{k-1})$, a lookup table for the log of the parameters of the Poisson distribution $\lambda_{z_k}$. We maintain a mean and a diagonal covariance matrix for the Gaussian distribution $p(x_i \mid z_k)$ for each $z_k$. For low-rank HSMMs (LHSMMs), we use the same parameterization for $p(z_k \mid z_{k-1})$ as in HMMs:

$$p(z_k \mid z_{k-1}) \propto \phi(\mathbf{u}_{z_{t-1}})^\top \phi(\mathbf{v}_{z_t}), \tag{13}$$

where $\mathbf{u}_z$ is the embedding of $z$ when $z$ is used as head, $\mathbf{v}_z$ its embedding when used as tail, and the feature map $\phi(x) = \exp(Wx)$. The emission parameterization is the same as in baseline HSMMs, a Gaussian kernel.

# G   Initialization and Optimization Hyperparameters

We initialize the parameters $W$, in $\phi(x) = \exp(Wx)$ and variants, of feature maps using orthogonal feature projections [Choromanski et al., 2020], and update it alongside the model parameters during training.

HMM parameters are initialized with the Xavier initialization [Glorot and Bengio, 2010].[15] We use the AdamW [Loshchilov and Hutter, 2017] optimizer with a learning rate of $0.001$, $\beta_1 = 0.9, \beta_2 = 0.999$, weight decay $0.01$, and a max grad norm of $5$. We use a state dropout rate of $0.1$, and additionally have a dropout rate of $0.1$ on the feature space of LHMMs. We train for 30 epochs with a max batch size of 256 tokens, and anneal the learning rate by dividing by 4 if the validation perplexity fails to improve after 4 evaluations. Evaluations are performed 4 times per epoch. The sentences, which

---

[14]We got rid of those constraints to allow for changing the total number of states, since otherwise we can't make any changes under a predefined state space.

[15]For banded experiments, we initialize the band parameters by additionally adding 30 to each element. Without this the band scores were too small compared to the exponentiated scores, and were ignored by the model.

we model independently from one another, are shuffled after every epoch. Batches of sentences are drawn from buckets containing sentences of similar lengths to minimize padding.

For the polyphonic music datasets, we use the same hyperparameters as the language modeling experiments, except a state dropout rate of 0.5 for JSB and Nottingham, 0.1 for Muse and Piano. We did not use feature space dropout in the LHMMs on the music datasets. For Nottingham and JSB, sentences were batched in length buckets, the same as language modeling. Due to memory constraints, Muse and Piano were processed using BPTT with a batch size of 8 for Muse and 2 for Piano, and a BPTT length of 128. We use $D = 256$ for all embeddings and MLPs on all datasets, except Piano, which due to its small size required $D = 64$ dimensional embeddings and MLPs.

For PCFGs, parameters are initialized with the Xavier uniform initialization [Glorot and Bengio, 2010]. We follow the experiment setting in Kim et al. [2019] and use the Adam [Kingma and Ba, 2017] optimizer with $\beta_1 = 0.75, \beta_2 = 0.999$, a max grad norm of 3, and we tune the learning rate from $\{0.001, 0.002\}$ using validation perplexity. We train for 15 epochs with a batch size of 4. The learning rate is not annealed over training, but a curriculum learning approach is applied where only sentences of at most length 30 are considered in the first epoch. In each of the following epochs, the longest length of sentences considered is increased by 1.

For HSMMs, we use the same initialization and optimization hyperparameters as Fried et al. [2020]: The Gaussian means and covariances are initialized with empirical means and covariances (the Gaussian parameters for all states are initialized the same way and they only diverge through training). The transition matrix is initialized to be uniform distribution for baseline HSMMs, and the transition embeddings are initialized using the Xavier initialization for LHSMMs. The log of Poisson parameters are initialized to be 0. We train all models for 4 epochs using the Adam optimizer with initial learning rate of 5e-3, and we reduce the learning rate 80% when log likelihood doesn't improve over the previous epoch. We clamp the learning rate to be at least 1e-4. We use a batch size of 5 following Fried et al. [2020], simulated by accumulating gradients under batch size 1 in order to scale up the number of states as much as we can. Gradient norms are clipped to be at most 10 before updating. Training take 1-2 days depending on the number of states and whether a low-rank constraint is used.

We use the following hardware for our experiments: for HMMs we run experiments on 8 Titan RTX GPUs with 24G of memory on an internal cluster. For PCFGs and HSMMs we run experiments on 1 Nvidia V100 GPU with 32G of memory on an internal cluster.

# H   HMM Rank Analysis

Table 4 contains the empirical ranks of trained HMMs and LHMMs, estimated by counting the number of singular values greater than 1e-5. Note that the feature dimension $N$ is the maximum attainable rank for the transition matrix of an LHMM. Although LHMMs often manage to achieve the same validation perplexity as HMMs at relatively small $N$, the ranks of the transition matrices are much lower than both their HMM counterparts as well as $N$. At larger state sizes, the ranks of learned matrices are almost half of their max achievable rank. Interestingly, this holds true for HMMs as well, with the empirical rank of the transition matrices significantly smaller than the number of states. Whether this implies that the models can be improved is left to future investigations.

# I   Low-rank and Banded HMM Parameterization

In some scenarios, the low-rank constraint may be too strong. For example, a low-rank model is unable to fit the identity matrix, which would have rank $L$. In order to overcome this limitation, we extend the low-rank model while preserving the computational complexity of inference. We perform experiments with an additional set of parameters $\theta \in \mathbb{R}^{L \times L}$ which allow the model to learn high-rank structure (the experimental results can be found in Tbl. 1). We constrain $\theta$ to have banded structure, such that $[\theta]_{z_{t-1}, z_t} = 0$ if $|z_t - z_{t-1}| > N/2$. See Fig. 3 for an illustration of banded structure.

Let band segment $B_z = \{z' : |z - z'| \leq N/2\}$. The transition probabilities are then given by

$$p(z_t \mid z_{t-1}) = \frac{[\theta]_{z_{t-1}, z_t} + \phi(\mathbf{u}_{z_{t-1}})^\top \phi(\mathbf{v}_{z_t})}{Z_{z_{t-1}}}, \tag{14}$$

| Model | $L$ | $N$ | rank($A$) | rank($O$) | Val PPL |
|-------|-----|-----|-----------|-----------|---------|
| HMM | 16384 | - | 9187 | 9107 | 144 |
| LHMM | 16384 | 8192 | 2572 | 7487 | 141 |
| LHMM | 16384 | 4096 | 2016 | 7139 | 144 |
| LHMM | 16384 | 2048 | 1559 | 6509 | 141 |
| LMM | 8192 | - | 5330 | 5349 | 152 |
| LHMM | 8192 | 4096 | 1604 | 5113 | 149 |
| LHMM | 8192 | 2048 | 1020 | 4980 | 153 |
| LHMM | 8192 | 1024 | 791 | 5033 | 161 |
| HMM | 4096 | - | 2992 | 3388 | 155 |
| LHMM | 4096 | 2048 | 1171 | 3300 | 154 |
| LHMM | 4096 | 1024 | 790 | 2940 | 156 |
| LHMM | 4096 | 512 | 507 | 3186 | 163 |

Table 4: Ranks and validation perplexities for HMMs and LHMMs. The number of states is given by $L$ and the dimensionality of the feature space by $N$. The HMM uses softmax for the emission, and therefore does not have a value for $N$. The transition matrix is denoted by $A$, and the emission matrix by $O$. The rank was estimated by counting the number of singular values greater than 1e-5. Models were trained with 0.1 state and feature dropout.

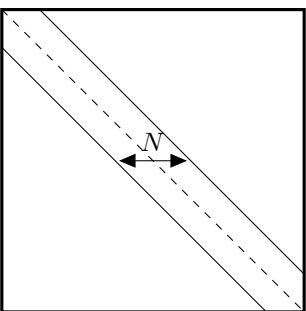

Figure 3: An example of a banded matrix with width $N$, which has $N/2$ nonzero elements on both sides of the diagonal for each row.

with normalizing constants

$$
\begin{aligned}
Z_{z_{t-1}} &= \sum_{z_t} [\theta]_{z_{t-1}, z_t} + \phi(\mathbf{u}_{z_{t-1}})^\top \phi(\mathbf{v}_{z_t}) \\
&= \sum_{z_t \in B_{z_{t-1}}} [\theta]_{z_{t-1}, z_t} + \phi(\mathbf{u}_{z_{t-1}})^\top \sum_{z_t} \phi(\mathbf{v}_{z_t}).
\end{aligned}
\tag{15}
$$

The normalization constant for each starting state $Z_{z_{t-1}}$ can be computed in time $O(N)$.

This allows us to perform inference quickly. We can use the above to rewrite the score matrix $\Psi_t \propto \theta + UV^\top$, which turns the inner loop of Eqn. 3 (specialized to HMMs) into

$$
\alpha_t = \Psi_t \beta_{t+1} \propto (\theta + UV^\top)\beta_{t+1} = \theta \beta_{t+1} + U(V^\top \beta_{t+1}),
\tag{16}
$$

omitting constants (i.e. emission probabilities and normalizing constants). Since $\theta$ is banded, the banded matrix-vector product $\theta \beta_t$ takes time $O(LN)$. This update, in combination with the low-rank product, takes $O(LN)$ time total. Each update in the hypergraph marginalization algorithm is now 3 matrix-vector products costing $O(LN)$ each, preserving the runtime of inference.

# J    Music Results

The full results on the polyphonic music modeling task can be found in Tbl. 5, with additional models for comparison. Aside from the RNN-NADE [Boulanger-Lewandowski et al., 2012], which

| Model | Nott | Piano | Muse | JSB |
|---|---|---|---|---|
| RNN-NADE | 2.31 | 7.05 | **5.6** | 5.19 |
| Seq-U-Net | 2.97 | **1.93** | 6.96 | 8.173 |
| R-Transformer | **2.24** | 7.44 | 7.00 | 8.26 |
| LSTM | 3.43 | 7.77 | 7.23 | 8.17 |
| STORN | 2.85 | 7.13 | 6.16 | 6.91 |
| LV-RNN | 2.72 | 7.61 | 6.89 | **3.99** |
| SRNN | 2.94 | 8.2 | 6.28 | 4.74 |
| DMM | 2.77 | 7.83 | 6.83 | 6.39 |
| LNF | 2.39 | 8.19 | 6.92 | 6.53 |
| TSBN | 3.67 | 7.89 | 6.81 | 7.48 |
| HMM | 2.43 | 8.51 | 7.34 | 5.74 |
| LHMM | 2.60 | 8.89 | 7.60 | 5.80 |

Table 5: Polyphonic music negative log-likelihood, measured in nats. The HMM models have $\mathcal{L} = 2^{11}$ states and the LHMM has rank $N = 2^9$, a 4:1 state:rank ratio.

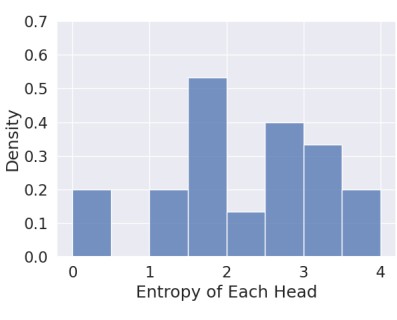

(a) Softmax Parameterization

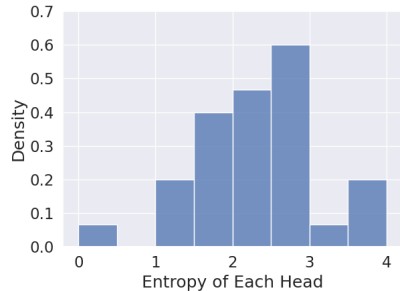

(b) Low-rank Parameterization

Figure 4: Histogram of entropies of $P(B\ C \mid A)$. The average entropy is 2.26 for softmax and 2.34 for the low-rank parameterization. We use $|\mathcal{N}| = 30$, $|\mathcal{P}| = 60$, and $N = 16$ for the rank.

models the full joint distribution of notes as well as temporal dependencies; autoregressive neural R-Transformer [Wang et al., 2019] (as reported by Song et al. [2019]) and LSTM (as reported by Ziegler and Rush [2019]); latent continuous LV-RNN [Gu et al., 2015] and SRNN [Fraccaro et al., 2016]; and latent discrete TSBN [Gan et al., 2015] and the baseline HMM; we additionally include the autoregressive Seq-U-Net Stoller et al. [2019], the continuous latent STORN [Bayer and Osendorfer, 2015], DMM [Krishnan et al., 2016] and LNF [Ziegler and Rush, 2019].

# K   PCFG Analysis

| Kernel for $B\ C$ | | | | PPL |
|---|---|---|---|---|
| $\mathcal{N} \times \mathcal{N}$ | $\mathcal{N} \times \mathcal{P}$ | $\mathcal{P} \times \mathcal{N}$ | $\mathcal{P} \times \mathcal{P}$ | |
| SM | SM | SM | SM | 243.19 |
| LR | SM | SM | SM | 242.72 |
| LR | LR | LR | SM | 259.05 |
| LR | LR | LR | LR | 278.60 |

Table 6: Model perplexities evaluated on the validation set of PTB. Here we use $|\mathcal{N}| = 30$, $|\mathcal{P}| = 60$, and $N = 16$ rank. SM denotes the use of softmax, while LR a low-rank factorization.

Figure 4 shows the entropy distribution of the production rules $H(P(B\ C|A))$ for both using softmax kernel and the approximation. The average entropies of the two distributions are close. Besides, under this setting, $P(B\ C \in \mathcal{N} \times \mathcal{N}|A)$ are close for both kernels as well (softmax 0.20, linear 0.21), eliminating the possibility that the kernel model simply learns to avoid using $B\ C \in \mathcal{N} \times \mathcal{N}$ (such as by using a right-branching tree).

In Table 6, we consider the effects of the mixed parameterization, i.e. of replacing the softmax parameterization with a low-rank parameterization. In particular, we consider different combinations of preterminal / nonterminal tails $B\ C \in \mathcal{N} \times \mathcal{N}$, $B\ C \in \mathcal{N} \times \mathcal{P}$, $B\ C \in \mathcal{P} \times \mathcal{N}$, and $B\ C \in \mathcal{P} \times \mathcal{P}$ (our main model only factorizes nonterminal / nonterminal tails). Table 6 shows that we get the best perplexity when we only use $K$ on $B\ C \in \mathcal{N} \times \mathcal{N}$, and use softmax kernel $K_{\text{SM}}$ for the rest of the space. This fits with previous observations that when the label space $|\mathcal{L}|$ is large, a model with a very small rank constraint hurts performance.[16]

# L    Speed and Accuracy Frontier Analysis

We provide plots of the speed and accuracy over a range of model sizes for HMMs and PCFGs, in Figure 5 (left and right respectively). Speed is measured in seconds per batch, and accuracy by perplexity. Lower is better for both.

For HMMs, we range over the number of labels $L \in \left\{ 2^{10}, 2^{11}, 2^{12}, 2^{13}, 2^{14} \right\}$. For softmax HMMs, more accurate models are slower, as shown in Figure 5 (left). However, we find that for any given accuracy for a softmax model, there exists a similarly accurate LHMM that outspeeds it. While we saw earlier in Figure 1 that at smaller sizes the low-rank constrain hurt accuracy, a model with a larger state size but lower rank achieves similar accuracy at better speed compared to a small HMM.

For PCFGs, we range over $L \in \{90, 180, 300\}$. We find a similar trend compared to HMMs: accuracy results in slower models, as shown in Figure 5 (right). However, the LPCFG does not dominate the frontier as it did with HMMs. We hypothesize that this is because of the small number of labels in the model. In the case of HMMs, smaller softmax HMMs were more accurate than the faster low-rank versions, but larger LHMMs with low rank were able to achieve similar perplexity at faster speeds. This may be realized by exploring LPCFGs with more state sizes, or simply by scaling further.

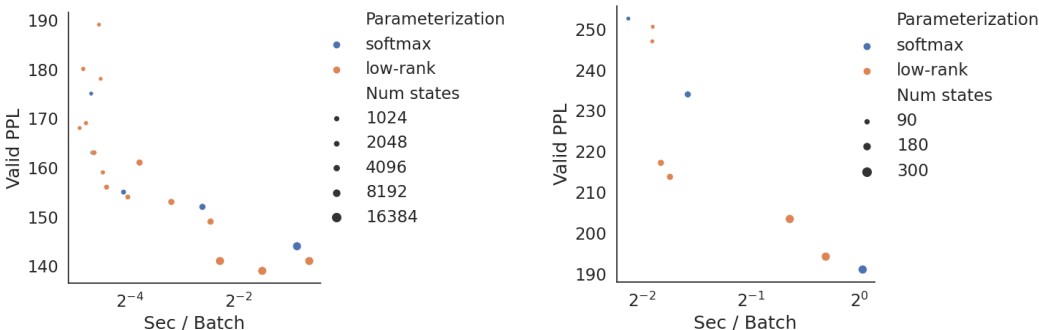

Figure 5: The speed, in seconds per batch, versus accuracy, in perplexity, for HMMs (left), PCFGs (right), and low-rank versions over a range of model sizes. As lower is better for both measures of speed and accuracy, the frontier is the bottom left.

# M    Potential Negative Impact

While work on interpretable and controllable models is a step towards machine that can more easily be understood by and interact with humans, introducing external-facing components leaves models possibly more vulnerable to adversarial attacks. In particular, the interpretations (in conjunction with

---

[16]In this particular ablation study, the size of $\mathcal{N} \times \mathcal{N}$ is only one-ninth of the total state space size $\{\mathcal{N} \cup \mathcal{P}\} \times \{\mathcal{N} \cup \mathcal{P}\}$.

the predictions) afforded by interpretable models may be attacked [Zhang et al., 2018]. Additionally, models with simple dependencies may be easier for adversaries to understand and then craft attacks for [Zhang et al., 2021, Liu et al., 2018].