# OpenReview forum: "Low-Rank Constraints for Fast Inference in Structured Models"
_NeurIPS.cc/2021/Conference — NeurIPS 2021 Poster_

### Official Review · Reviewer_Ys2d · 2021-07-06

**Rating:** 6
**Confidence:** 5

**Summary:**

A variety of structured probabilistic models (HMMs, PCFGS, etc) permit inference using dynamic programming. A primary driver of computational cost is the matrix multiplications required for passing the messages in the associated hyper-graph. This paper proposes using low-rank matrices for the costs in these hyper-edges, which allows for cost that scales with the rank of the matrix. Since this is rank is a free parameter, it can be used to tune a speed-accuracy tradeoff.

**Limitations And Societal Impact:**

Yes

**Main Review:**

**Expressivity of the model**
Can you formally argue why using a model with high-dimensional hidden states, but a low-dimensional kernel, is not equivalent to a model with lower-dimensional hidden states? It's definitely not clear why the machinery of this paper is necessary, when the hidden state size is a hyperparameter that the user could decrease if computational complexity was an issue. If they're not formally equivalent, they still strike me as similar in terms of expressivity (which is supported by your experiments).


**Training**
There are far too few details regarding how the models are trained. I'm assuming you used EM. This is very dependent on initialization in practice. How did you do it?


**Computational Complexity**
With modern hardware, big-O analysis may not be helpful because certain operations can be done in parallel. For example, for O(L N^2) inference, the cost is sequential in L, but parallelizable in N. Also, for certain ranges of N, the cost of matrix multiplication on a GPU is more or less constant.


The right panel of fig 1 should be in log space. Does it give the slope we expect?

**Results**
I expected to see much more of an analysis of an accuracy-speed tradeoff. However, besides the HMM experiments, all other results only consider accuracy. In terms of accuracy, I could not find a convincing result demonstrating that using LHMM vs. HMM or LPFCG vs. PCFG gave a meaningful performance improvement.


Overall, I am confused why much of the analysis sweeps over the number of hidden states. Shouldn't this just be a hyper-parameter that gets tuned for each model? For example, figure 1 would be much more clear if it was collapsed into a single scatter plot of PPL vs. speed.


**Related Work**
I found a few important related papers. Can you comment on the relationship between your proposed method and these works?

Thai et al., "Embedded-State Latent Conditional Random Fields for Sequence Labeling"

McAuley et al., "Exploiting Data-Independence for Fast Belief-Propagation"

Belanger et al., "MAP Inference in Chains using Column Generation"


**After authors' response**
Score raised from weak reject to weak accept.

**Time Spent Reviewing:**

1

---

> ### Author Response · Authors · 2021-08-11
> **Re: Official Review of Paper11371 by Reviewer Ys2d**
>
> Thank you for your thoughtful feedback, we provide responses below.
>
> **Expressivity**: A common misconception is that a rank-N HMM can be converted to an N-state HMM, but this is not true. The intuition is that even with a low-rank transition matrix, the larger number of states enable a more expressive emission: for example, in an HMM with $L$ states but a rank-$N$ transition matrix, the emission matrix can have a higher rank compared to an HMM with $N$ states when the number of emissions is higher than the number of states (which is usually the case in practice). This is shown empirically in Table 4 of Appendix E.  In fact, Siddiqi et al 2010 [rrhmm] stated “So, rank-k RR-HMMs (which can have m >> k states) can model sets of predictive distributions which k-state HMMs cannot.” (Section 2.1, last paragraph). Our response to reviewer HMa8 provides a more detailed example.
>
> **Training**: We train by directly optimizing the exact log marginal likelihood with stochastic gradient ascent. The log marginals are computed via dynamic programming, e.g. the forward algorithm and inside algorithm for HMMs and PCFGs.
>
> **Sensitivity to initialization**: While optimizing the log marginal likelihood is sensitive to initialization (as are all models with nonconvex objectives), prior work has found that neural parameterizations work well empirically [cpcfg].
>
> **Computational complexity**: This is a good point. First, we will clarify that our approach has an extra factor of 2 hidden under the big-O notation (due to having 2 smaller matrix multiplications of $O(LN)$ instead of 1 bigger matrix multiplication of $O(L^2)$). While in theory the $L^2$ complexity can be parallelized with an infinite number of GPU cores, in practice $L^2$ is larger than the number of GPU cores, and we empirically observe speedups by using smaller matrix multiplications which reduces a factor of $L$ to a factor of $N$ (Fig 1 right). Besides, another practical benefit of having smaller matrix multiplications is that it takes less GPU memory and allows scaling to larger state sizes with better accuracies (as shown in Table 2 right).
>
> **The right panel of figure 1 should be log space**: We will include the log-log version of Fig. 1 in the next version.
>
> **Does it give the slope we expected**: we can’t post the log-log figure here, but the slope is as expected, slowing down slightly more than 2x as the state size doubles. The initial slope (from 2^10 to 2^12 states) appears to be unaffected by the low-rank factorization because the runtime is dominated by other parts of the model.
>
> **Collapsing into a single PPL vs speed**: Fig 1 demonstrates that LHMMs empirically obtain similar performance to HMMs while achieving speedups. We will make this clearer by including a single plot of PPL vs speed, as suggested.
>
> **Sweeps over the number of hidden states**: prior works [scalehmm] have shown that increasing the model size leads to increased performances when we use early stopping, so we should expect a better accuracy with a larger number of hidden states. The only downside of using a larger model size is increased computational costs, so our plots can be understood as showing this tradeoff between accuracy and computational cost.
>
> **Speed for other models**: our cluster is down for maintenance this week, but we will post the numbers when it’s back.
>
> **Related work**:
>
> Thank you for the references! We will discuss these in our next version.
>
> - Thai et al., "Embedded-State Latent Conditional Random Fields for Sequence Labeling"
>
> The low-rank factorization here is identical to the one we use for speeding up inference. However, the low-rank factorization was chosen here for regularization purposes (i.e. reducing the number of parameters), whereas we emphasize the computational gains.
>
> - McAuley et al., "Exploiting Data-Independence for Fast Belief-Propagation"
>
> This paper proposes a method for speeding up MAP inference without introducing model constraints. We focus on marginal inference, and introduce a low-rank constraint on key subsets of model parameters. Due to our focus on marginal inference, our technique combines naturally with automatic differentiation and is easily applied to models with neural components, which would be more difficult with MAP EM.
>
> - Belanger et al., "MAP Inference in Chains using Column Generation"
>
> Similarly to the above paper, this paper relies on the sparsity of MAP inference for speedups. This approach is able to prune configurations, which is not viable in marginal inference due to the smoothed variational objective. This approach also does not introduce model constraints.
>
> **References**
>
> [cpcfg] Yoon Kim, Chris Dyer, and Alexander Rush. 2019. Compound probabilistic context-free grammars for grammar induction. In Proceedings of the 57th Annual Meeting of the Association for Computational Linguistics, pages 2369–2385, Florence, Italy. Association for Computational Linguistics.
>
> [rrhmm] S. M. Siddiqi, B. Boots, and G. J. Gordon. Reduced-rank hidden Markov models. In Thirteenth International Conference on Artificial Intelligence and Statistics, 2010.
>
> [scalehmm] Justin Chiu and Alexander Rush. 2020. Scaling hidden Markov language models. In Proceedings of the 2020 Conference on Empirical Methods in Natural Language Processing (EMNLP), pages 1341–1349, Online. Association for Computational Linguistics.

---

> > ### Author Response · Authors · 2021-08-14
> > **Speed for Other Models**
> >
> > PCFG Speed Numbers (measured during training):
> >
> > \\begin{array} {|r|r|r|r|r|r|r|}\\hline
> > |\mathcal{N}| & |\mathcal{P}| & Model & N &  PPL & Batch/Second \\\\\\hline
> > 30  & 60    & PCFG & - & 252.60 & 4.37 \\\\
> >     &       & LPCFG & 8 &  247.02    & 3.75     \\\\
> >     &       & LPCFG & 16 & 250.59    & 3.74   \\\\
> > \\hline
> > 60  & 120   & PCFG & - & 234.01 & 2.99\\\\
> >     &       & LPCFG & 16& 217.24 & 3.55 \\\\
> >     &       & LPCFG & 32& 213.81 & 3.35\\\\
> > \\hline
> > 100 & 200   & PCFG & - &  191.08   & 0.98 \\\\
> >     &       & LPCFG & 32& 203.47 & 1.56 \\\\
> >     &       & LPCFG & 64& 194.25 & 1.24 \\\\\hline
> > \\end{array}
> >
> > LPCFG is faster than PCFG for the configuration $|\mathcal{N}|=60$ nonterminals, $|\mathcal{P}|=120$ preterminals and $|\mathcal{N}|=100$, $|\mathcal{P}|=200$. Note that LPCFG is faster even when the number of features $N>\frac{|\mathcal{N}|}{2}$ (such as $|\mathcal{N}|=60$ and $N=32$), because different from the HMM case, here the linearized kernel approach does not have an extra factor of 2 due to the score matrix not being square: we are essentially replacing the multiplication of a matrix of size $(T^2,|\mathcal{N}|^2)$ with a matrix of size $(|\mathcal{N}|^2, |\mathcal{N}|)$ (complexity $T^2|\mathcal{N}|^3$) by two separate (smaller) matrix multiplications:
> >
> > 1. the multiplication of a matrix of size $(T^2,|\mathcal{N}|^2)$ with a matrix of size $(|\mathcal{N}|^2, N)$ (complexity $T^2|\mathcal{N}|^2N$);
> > 2. the multiplication of a matrix of size $(T^2,N)$ with a matrix of size $(N, |\mathcal{N}|)$ (complexity $T^2N|\mathcal{N}|$).
> >
> > Note that here the first step dominates the complexity of the second step as $|\mathcal{N}|^2>>|\mathcal{N}|$, unlike the HMM case where these two steps have the same complexity due to having a square scoring matrix.
> >
> >
> > HSMM speed numbers (measured during training):
> >
> > \\begin{array} {|r|r|r|r|r|r|r|}\\hline
> > Model & L & N & NLL & Batch/Second \\\\
> > \\hline
> > HSMM & 2^6 & - & 1.428\times 10^5 & 1.28 \\\\
> > HSMM & 2^7 & - & 1.427\times 10^5  & 0.45\\\\
> > HSMM & 2^8 & - & 1.426\times 10^5 & 0.13 \\\\
> > \\hline
> > LHSMM & 2^7 & 2^7 & 1.427\times 10^5 & 0.24 \\\\
> > LHSMM & 2^8 & 2^6 & 1.426\times 10^5 & 0.20 \\\\
> > LHSMM & 2^9 & 2^5 & 1.424\times 10^5 & 0.18 \\\\
> > LHSMM & 2^{10} & 2^4 & 1.423\times 10^5 & 0.10 \\\\
> > \\hline
> > \\end{array}
> >
> > When the number of states $L=2^7$, the number of features $N=2^7$, LHSMM is slower than HSMM, which is not surprising given that $N=L$. However, when $L=2^8$, $N=2^7$, LHSMM is faster than HSMM ($\times1.53$). When $L\ge2^9$, HSMM would throw out-of-memory errors, but LHSMM can still train if we halve $N$ as we double $L$ (note that even LHSMM with $L=2^9$ and $N=2^5$ is faster than HSMM with $L=2^8$), and the best log-likelihood is achieved at $L=2^{10}$ and $N=2^4$.

---

> > > ### Comment · Reviewer_Ys2d · 2021-08-17
> > > **thanks for the detailed authors' response**
> > >
> > > I found that your response (both to my comments and to other reviewers) adequately addressed my questions, particularly regarding the relationship between a low-rank HMM and an HMM with fewer hidden states. I've increased my score. I strongly encourage you to update the paper to include details about the expressivity of a low-rank HMM. Your comment to the first reviewer should be a lemma in the paper.

---

### Official Review · Reviewer_51Nb · 2021-07-13

**Rating:** 6
**Confidence:** 4

**Summary:**

Structured models e.g. HMM are shown to have good performance when the number of labels is very large. As one of the main computational bottlenecks of this is matrix-vector multiplications, the paper proposes to use low-rank decomposition to reduce the matrix-vector multiplication cost. The key challenge is which kernel to use for the low-rank decomposition so that the property of the transition matrix still holds.

The paper presents several experiments, cover several fields applications (language modelling, music modelling, video modelling) and structured architectures (HMM, CFG, HSMM). The proposed low-rank decomposition is shown to help reduce computational cost (2 to 2^6 times) while losing little or no accuracy.

Contributions:
- the paper proposes low-rank decomposition for matrix-vector multiplication to reduce the computational cost of structured architecture when the number of labels is very large,
- the proposed low-rank decomposition is applicable to any structured architectures whose a bottleneck is matrix-vector multiplication


**Limitations And Societal Impact:**

The submission includes "potential negative impact" in an appendix.

**Main Review:**

**Originality**

As low-rank decomposition proposed in this paper is very much similar to e.g. Choromanski et al (2020), it is hard to say that the work is original. The main difference of the paper is that the authors apply similar decomposition technique to structured models, with thoughtful arguments (e.g. how to apply to HMM, CRF) and experiments.

**Quality**

As structured models (e.g. HMM, CFG) and low-rank decomposition similar in this paper are well studied, the paper is technically sound. The experiment results confirm that the proposed low-rank decomposition work as expected: the computational time is reduced about 8 times (depending on settings) whereas the accuracy lost is small.

**Clarity**

The paper is very well written, with lots of details, thorough experiments. I particular like the flow of the paper, showing the clear motivation: from how to use matrix language describing structured models to the use of kernels.

There are some points that I didn't catch:
1. Line 145: what is the role of  \alpha_v here?
2. The authors drop the normalising constants c after the equation line 140. It is unclear how these factors are learnt.
3. In the experiments, N has different ranges for different experiments (line 219 - 226). This leads to one more hyper param to tune. I'm wondering whether it's a problem, or whether the authors have a "universal" wisdom to choose N.
4. In practise, is the computation time reduction worth? For instance, looking at table 1, AWD-LSTM significantly outperforms LHMM (57.3 vs 131.8 ppl). If we make AWD-LSTM to have similar running time like LHMM (e.g. reduce its #parameters), will be AWD-LSTM outperformed by LHMM?

The most crucial thing that I hardly get is the title of the paper "linearized structured models". Although I see where "linearize" comes from, this title is misleading as it might give a reader (e.g. me) an intuition that the paper proposes a method to turn nonlinear models to linear model.

**Significance**

This paper proposes an interesting idea for reducing theoretical computational cost. This theoretical result is important, though it is hard to see whether the empirical result is.


All in all I like the paper, but its originality is questionable.

**Time Spent Reviewing:**

7h

---

> ### Author Response · Authors · 2021-08-11
> **Re: Official Review of Paper11371 by Reviewer 51Nb**
>
> Thank you for your thoughtful feedback, we provide responses below.
>
> **Line 145**: Thanks for catching this, $\alpha_u$ being here is a mistake. We will also add the missing dimension of $\beta_v$.
>
> **Normalizing constants**: Those are calculated from $U_eV_e^\top$ and then we divied  $U_eV_e^\top$ by them to get a normalized $\Phi$. Note that these normalizing constants can be computed as an efficient series of matrix-vector products in time O(features * labels) as follows: $c_v = [U_e(V_e^\top 1)]_v$. We can then use these normalizing constants in Algorithm 3.
>
> **Choice of N**: The best choice for N would have to be determined empirically based on the application. We target parameter matrices that bottleneck computation, but also happen to be very important for expressivity, resulting in a sensitive speed-expressivity tradeoff. In general, since we are replacing one big matrix-vector product with two smaller ones, we need N to be less than half of L to get a speedup (we can get a speedup with a larger N in the case of PCFGs since the scoring matrix there is rectangular).
>
> **Comparison to LSTM**: Unfortunately, prior work shows that HMMs are handily outperformed by LSTM language models [scalehmm]. However, the same work shows that HMM performance does improve with size. Our focus is on improving the speed of inference for a given size, with the hope of enabling larger models.
>
> **Title**: This is a good point, and we will consider changing the title to “Low-rank Factorizations for Fast Inference in Structured Models.”
>
> **References**
>
> [scalehmm] Justin Chiu and Alexander Rush. 2020. Scaling hidden Markov language models. In Proceedings of the 2020 Conference on Empirical Methods in Natural Language Processing (EMNLP), pages 1341–1349, Online. Association for Computational Linguistics.

---

> > ### Comment · Reviewer_51Nb · 2021-08-21
> > **thanks the authors for the response**
> >
> > I would like to thank the authors for the response. I found this paper nice but I'm still skeptical about its originality. I thus keep my score.

---

### Official Review · Reviewer_Yt4e · 2021-07-14

**Rating:** 7
**Confidence:** 4

**Summary:**

The paper proposes a way of speeding up inference in directed graphical models by noting that the most expensive operation in the inner loop can be represented as a matrix-vector product and that if the model is parametrized in a different way this matrix-vector product can be done in lower computational complexity than in the usual algorithms. The interesting experimental result is that many classic models (including HMMs and CFG parsing) in reasonable applications can be parametrized in this way without a significant loss in model quality but a significant gain in speed, which allows for much larger model capacity than previously considered in the literature, leading to higher model quality at a fixed computational budget.

**Limitations And Societal Impact:**

No issues here.

**Main Review:**

Overall the paper is well written and clear, and the improvements are compelling.

I wish the fully instantiated linear algorithms were presented in the same notation as algorithm 2 (the reference algorithms expressed in hypergraph terminology) as that would make it easier to understand particular details of the linear parametrization used in the paper.

I also wish the linear algebra was expressed directly as well as being coated in kernel language. AFAICT this is a fundamental property of the fact that the matrix of probabilities might be expressible in a factorized form; this allows for the HMM case to reparametrize a multiplication of the form [labels, labels] [labels] with complexity O(labels^2) with a multiplication of the form [labels, features] [features, labels] [labels] which can be computed in O(features * labels) if the multiplication is associated to the right (i.e. [labels, features] ([features, labels] [labels])).

One thing which is not obvious to me from reading the paper is how does this ensure that the probability distribution of the PCFGs/HMMs remains normalized, while with a softmax parametrization it's trivial to ensure that every conditional probability is normalized. This would also be improved by more details on the particular algorithm used.

Clarifying the linear algebra structure would also help clarify generalizations to other semirings (like the max-product ring useful when doing maximum a posteriori inference and where the normalization of the conditional probabilities is immaterial).

I believe the paper needs to be revised to more explicitly clarify the issues raised above.

Minor nits:
 line 46, "attention" is repeated


---

Note: score revised after author response.

**Time Spent Reviewing:**

2

---

> ### Author Response · Authors · 2021-08-11
> **Re: Official Review of Paper11371 by Reviewer Yt4e**
>
> Thank you for your thoughtful feedback, we provide responses below.
>
> **Linearized algorithms for HMM and PCFG**: This is a great idea, and we will include this in our next version.
>
> **Removing kernel framing**: You are correct that the low-rank factorization is simply a constraint that is orthogonal to the particular kernel parameterization and feature map. We will simplify the presentation by de-emphasizing the kernel perspective.
>
> **Normalizing constants**: After applying a low-rank factorization, the normalizing constants for messages can be computed as an efficient series of matrix-vector products in time O(features * labels) as follows: $c_v = [U_e(V_e^\top 1)]_v$. We can then use these normalizing constants in Algorithm 3.
>
> **Generalizations to other semirings**: The key operation the low-rank constraint affects is the matrix-vector product, which can be expressed straightforwardly in either the standard or log semirings, allowing for efficient marginal inference. The low-rank constraint does not translate easily to the max-plus or max-times semirings.

---

> > ### Comment · Reviewer_Yt4e · 2021-08-11
> > **Response to comments**
> >
> > Given the author response I'll edit my review scores to be more positive. I also think the discussion of other semirings belongs in the paper as at least a footnote.

---

### Official Review · Reviewer_HMa8 · 2021-07-17

**Rating:** 5
**Confidence:** 4

**Summary:**

The paper proposes an approach to reduce the cost of dynamic programming inference in structured discrete models such as HMMs and PCFGs. Viewing each message passing step as a multiplication by a matrix of conditional scores, the proposed approach replaces this matrix by a low-rank matrix, parameterized as the linearization of a finite-dimensional kernel evaluated on state embeddings. This speeds up inference by roughly a factor of `L / N`, where L is the number of states and N is the rank of the kernel. The method is evaluated for HMMs on tasks of modeling language and polyphonic music, for PCFG parsing on the Penn Treebank, and for a video modeling task with HSMMs.

**Limitations And Societal Impact:**

The limitations of the method (e.g., the restrictions on parameterization) are clearly discussed and examined experimentally, which is admirable.

The work is on core algorithms and raises no specific societal impact concerns.

**Main Review:**

The paper is overall well written and straightforward to follow. I quite liked the unified exposition of dynamic programming algorithms as matrix-vector multiplication on hypergraphs. These algorithms are fundamental even in the modern 'neural' paradigm, and a convincing asymptotic speedup would be a significant contribution.

Unfortunately, I don't think the abstract's claim that "we can reduce the complexity of inference by a factor of the hidden state size" is supported by the experiments. Given that claim, I would expect to see LHMM experiments using a factor of N more states than the largest practical HMMs, but the experiments explore only relatively small constant factors (2:1, 4:1, etc, maxing out around 32:1), and even then struggle to demonstrate practical gains. That said, I do applaud the both range of experiments and the inclusion of relevant baselines.

My fundamental issue with the story of this paper is that I don't have any intuition (and the paper doesn't seem to provide any) for why an LHMM with L states and rank N < L should be a different model class from just learning an HMM on N states, which is even cheaper (and the analogous statement for PCFGs). Given a rank-N transition matrix, you can always derive an equivalent HMM on a set of N 'abstract' states $w_t \in Z_N$, with transition probabilities $$p(w_{t+1} | w_t) = \sum_z p(w_{t+1} | Z_t=z) p(Z_t =z| w_t) = (V_{e_1})^T_{w_{t + 1}} (U_{e_2})_{w_t}$$
and emission model
$$p(x_t | w_t) = \sum_z p(x_t | Z_t=z)p(Z_t = z | w_t)$$
derived by marginalizing out the L-dimensional state vectors. The 2009 Siddiqi et al. work on low-rank HMMs explicitly acknowledges this (the abstract mentions that 'the dynamics evolve in a k-dimensional subspace') but focuses on the case with Gaussian emissions, where you do at least end up with more expressivity from implicitly emitting a mixture of L rather than N Gaussians (this would explain the improvement of LHSMM over HSMM for the video modeling task), but in the discrete case I'm not sure why you wouldn't just use the smaller model. Can the authors comment? I could imagine that overparameterizing the model might be helpful for optimization, or that the ability to also add banded structure makes the higher-rank model more interesting, but I'd want to see those stories developed and supported by evidence. Either way, I think this equivalence is worth mentioning.

As a somewhat more superficial quibble, I found the framing in terms of kernels a bit confusing. The 'point' of kernel methods is generally that you can use the dual representation (or approximations like random features) to work in high-dimensional spaces without paying the corresponding costs. But if I understand the proposed approach, it's not attempting to linearize any particular high-dimensional computation; rather, it's explicitly learning a low-dimensional feature map. It seems like the main idea of the paper is essentially 'parameterize the score matrix as the product of low-dimensional embedding vectors learned by gradient descent'. That's a reasonable idea, but cloaking it in the language of kernels seems unnecessarily obfuscatory when it could equally well be explained with undergrad linear algebra (which is not a flaw! simple approaches are better, if they work well).

I imagine that a lot was learned in the course of writing this paper, but in its current form I unfortunately don't think it rises past the NeurIPS bar. If I'm significantly misunderstanding the approach, that would of course change my view. At the least, I'd like to see the claims of order-of-magnitude improvement either scaled back, or supported by stronger evidence.

****************************
Update: Thanks to the authors for their response: as they point out, low-rank HMMs are more expressive than reduced-state-size HMMs even in the discrete case once you consider the joint distribution of $x_t$'s across timesteps (vs in the continuous case, where implicitly using a mixture of $L > N$ Gaussians would get you more expressivity even at a single timestep). This addresses my deepest concern with the paper, and I've increased my score by a point. I think this could be a solid paper with the promised revisions, though unfortunately can't fully credit them sight unseen.

**Time Spent Reviewing:**

4

---

> ### Author Response · Authors · 2021-08-11
> **Re: Official Review of Paper11371 by Reviewer HMa8**
>
> Thank you for your thoughtful feedback, we provide responses below.
>
> **Claim in abstract**: The abstract is indeed not supported by the results, as we find experimentally that an effective rank is highly dependent on the state size. We will change this to "empirical speedups," and apologize for the error.
>
> **Relationship between low-rank HMMs and smaller state HMMs**: even in the discrete case, an HMM with a transition matrix of rank N cannot always be converted to an equivalent HMM with N states: while a rank-N HMM can be converted to an N-dimensional PSR (Predictive State Representations), PSRs are a larger class than HMMs (the states of PSRs are N-dimensional continuous vectors). In fact, Siddiqi et al 2010 [rrhmm] mentioned “So, rank-k RR-HMMs (which can have m >> k states) can model sets of predictive distributions which k-state HMMs cannot.” (Section 2.1, last paragraph, note that this statement does not specify whether it's discrete or continuous).
>
> We can use a concrete example to show that there are distributions that rank-N HMMs can model but N-state HMMs cannot: let sequence length $T=2$, rank $N=2$, observations $x_t\\in\\{0, 1, 2\\}$.
>
> 1. Specification of a rank-2 HMM with 3 states ($z_t\\in \\{0,1,2\\}$):
>
> transition probabilities (rows $z_t$, columns $z_{t+1}$): $P(z_{t+1} |z_{t}) = \\begin{bmatrix}\\frac{1}{3} & \\frac{1}{3} & \\frac{1}{3} \\\\ 0 & 1 & 0 \\\\ \\frac{1}{2} & 0 & \\frac{1}{2} \\end{bmatrix} =  \\begin{bmatrix}\\frac{1}{3} & \\frac{2}{3} \\\\ 1 & 0 \\\\  0 & 1 \\end{bmatrix}   \\begin{bmatrix}0 & 1 & 0 \\\\ \\frac{1}{2} & 0 & \\frac{1}{2}  \\end{bmatrix} = UV^T$
>
> emission probabilities (rows $z_t$, columns $x_t$): $P(x_t|z_t) = \\begin{bmatrix}1 &0 & 0 \\\\ 0 & 1 & 0 \\\\ 0 & 0 &1\\end{bmatrix}$
>
> initial auxiliary state: $z_0 = 0$, such that the distribution of the first state $P(z_1) =  \\begin{bmatrix}\\frac{1}{3} & \\frac{1}{3} & \\frac{1}{3}\\end{bmatrix}$
>
> Then the marginal distribution (row $x_1$, column $x_2$): $P(x_1, x_2) = \\begin{bmatrix}\\frac{1}{9} & \\frac{1}{9} & \\frac{1}{9} \\\\ 0 & \\frac{1}{3} & 0 \\\\ \\frac{1}{6} & 0 & \\frac{1}{6} \\end{bmatrix}$.
>
> We will show in the following that this marginal is not achievable under a 2-state HMM. The intuition is that the second row ($P(x_2|x_1=1)= \\begin{bmatrix}0& 1 & 0\\end{bmatrix}$) would need one state $z_2=a$ to model (and this state will only emit $x_t=1$, and the third row ($P(x_2|x_1=2)= \\begin{bmatrix}0.5& 0 & 0.5\\end{bmatrix}$) would need the other state $z_2=b\ne a$ to model (since $a$ cannot emit $x_t=0$). We know that $z_1=b$ for $x_1=2$ since $z_1=a$ can only emit $x_1=1$, then the last row also tells us that $z_1=b$ cannot transition to $z_2=a$, since otherwise $P(x_2=1|x_1=2)$ would be nonzero (because $z_2=a$ can emit $x_2=1$). Now for the first row ($P(x_2|x_1=0)$), the state $z_1$ must be $b$ since $z_1=a$ cannot emit $x_1=0$, so $z_2$ must also be $b$ because $b$ cannot transit to $a$. Then $P(x_2=1|x_1=0)$ would also be 0 since $b$ cannot emit 1, contradicting the given marginals.
>
> 2. A more rigorous proof.
> There does not exist any 2-state HMM with the above marginal distribution. To prove this, we will first show that there is only one valid emission distribution in a 2-state HMM, and then derive a contradiction when we look at transitions.
>
> We will use the equations $P(x_1, x_2) = \\sum_{z_2}\\left[\sum_{z_1}P(z_1)P(x_1|z_1)P(z_2|z_1)\\right]P(x_2|z_2)$. We denote $\sum_{z_1}P(z_1)P(x_1|z_1)P(z_2|z_1)$ as $f(x_1,z_2)$, then we can write those equations in the matrix form:
>
> $ \\begin{bmatrix} f(0, 0) & f(0, 1) \\\\ f(1, 0) & f(1, 1) \\\\ f(2, 0) & f(2, 1) \\end{bmatrix} \\begin{bmatrix} P(x_2|z_2=0) \\\\ P(x_2|z_2=1) \\end{bmatrix} =P(x_1, x_2) =  \\begin{bmatrix}\\frac{1}{9} & \\frac{1}{9} & \\frac{1}{9} \\\\ 0 & \\frac{1}{3} & 0 \\\\ \\frac{1}{6} & 0 & \\frac{1}{6} \\end{bmatrix}$
>
> Looking at the second row of this equation: $f(1,0) P(x_2|z_2=0) + f(1,1)P(x_2|z_2=1)= \\begin{bmatrix}0 & \\frac{1}{3} & 0 \\end{bmatrix}$, $f(1,0)$ and $f(1,1)$ can't be both 0 since otherwise the result would be a zero vector. Without loss of generality, assume $f(1,0)\ne 0$, then $P(x_2=0|z_2=0)=P(x_2=2|z_2=0)=0$, since all terms are non-negative and if any of them is greater than
> 0, then the result would be greater than 0 at the corresponding position. Therefore, $P(x_2|z_2=0)=\\begin{bmatrix}0 & 1& 0 \\end{bmatrix}$.
>
> Now let's look at the last row of the equation: $f(2,0) P(x_2|z_2=0) + f(2,1)P(x_2|z_2=1)= \\begin{bmatrix} \\frac{1}{6} &0 &  \\frac{1}{6} \\end{bmatrix}=P(x_2|x_1=2)$. Since $P(x_2|z_2=0)=\\begin{bmatrix}0 & 1& 0 \\end{bmatrix}$, $f(2,0)$ must be 0, otherwise the result would have a non-zero $P(x_2=1|x_1=2)$. Therefore, $f(2,1)P(x_2|z_2=1)= \\begin{bmatrix} \\frac{1}{6} &0 &  \\frac{1}{6} \\end{bmatrix}$, hence $P(x_2|z_2=1)= \\begin{bmatrix} \\frac{1}{2} &0 &  \\frac{1}{2} \\end{bmatrix}$.
>
> Putting the above two paragraphs together, we have the full emission matrix now: $P(x_t|z_t)=\\begin{bmatrix}0 & 1& 0\\\\  \\frac{1}{2} &0 &  \\frac{1}{2}  \\end{bmatrix}$.
>
> Using this emission matrix, we can find deterministic posterior mappings from observed variables $x_1$ to latent variables $z_1$. First, since $P(x_1=1|z_1=1)=0$ , we have $P(z_1=1|x_1=1)=0$ using Bayes’ rule, so $P(z_1=0|x_1=1)=1$. Similarly, we can prove that $P(z_1=1|x_1=0)=1$ and $P(z_1=1|x_1=2)=1$.
>
> Now we can find a contradiction when we consider transitions from $z_1$ to $z_2$: first we will prove that $z_1=1$ cannot transit to $z_2=0$ and can only transit to $z_2=1$:  $P(x_2=1|x_1=2)\ge P(x_2=1, z_2=0, z_1=1|x_1=2) = P(x_2=1|z_2=0)P(z_2=0|z_1=1)P(z_1=1|x_1=2)$, substituting  $P(z_1=1|x_1=2)=1$ and $P(x_2=1|z_2=0)=1$ into this inequality, we have $P(x_2=1|x_1=2)\ge P(z_2=0|z_1=1)$. Since $P(x_2=1|x_1=2)=0$, we have $P(z_2=0|z_1=1)=0$. Then we will show that $P(x_2=1|x_1=0)$ must be 0 since $z_1|x_1=0$ has to be 1 and it can only transit to $z_2=1$ which cannot emit $x_2=1$:  $P(x_2=1|x_1=0) = \sum_{z_1, z_2}P(x_2=1, z_1, z_2|x_1=0) =  \sum_{z_1, z_2}P(x_2=1 | z_2)P(z_2|z_1) P(z_1|x_1=0)$. Since $P(z_1=0|x_1=0)=0$, $P(x_2=1|x_1=0) =\sum_{z_2} P(x_2=1|z_2)P(z_2|z_1=1)P(z_1=1|x_1=0)$. Since $P(z_2=0|z_1=1)=0$, this sum can further reduce to $P(x_2=1|x_1=0) =P(x_2=1|z_2=1)P(z_2=1|z_1=1)P(z_1=1|x_1=0)$. Using the emission matrix, $P(x_2=1|z_2=1)=0$, so $P(x_2=1|x_1=0) =0$, which contradicts the desired marginal $P(x_2=1|x_1=0) =\\frac{1}{3}$. Therefore, we have completed the proof that it is impossible to use an HMM with 2 states to get the same marginal distribution.
>
> **Removing the kernel framing**: This is a great point, and we will de-emphasize connections to kernels to simplify exposition.
>
> **References**
>
> [rrhmm] S. M. Siddiqi, B. Boots, and G. J. Gordon. Reduced-rank hidden Markov models. In Thirteenth International Conference on Artificial Intelligence and Statistics, 2010.

---

### Decision · Program_Chairs · 2021-09-28

**Decision:**

Accept (Poster)

**Comment:**

This paper analyzes the speed up in reduced-rank forms of HMMs and PCFGs. The reviewers agree that the paper is well written, has useful contributions, and the experiments are good. I strongly suggest following up on the change of title, including the low-rank HMM discussion, and the semi-ring discussions from the rebuttal process in the paper.

**Consistency Experiment:**

NeurIPS has a long history of experimentation. In 2014, NeurIPS ran an experiment in which 10% of submissions were reviewed by two independent committees to quantify the randomness in the review process. This year, we repeated a variant of this experiment to see how the quality of the review process has changed over time.  This paper was part of the experiment and was therefore assigned to two committees (consisting of reviewers, an Area Chair, and a Senior Area Chair) that reached independent decisions.  If both committees made the same recommendation, this recommendation was followed. If a single committee recommended acceptance, the paper was accepted (with the exception of a few cases in which the other committee identified what we considered a fatal flaw, e.g., an error in a key result).

This copy’s committee reached the following decision: **Accept (Poster)**

The other committee assigned to the paper recommended **Reject**.  You can find the other set of reviews, along with any follow up discussion with the authors here:
https://openreview.net/forum?id=Mcldz4OJ6QB